



# Microbial community structure in the Western Tropical South Pacific

Nicholas Bock[1], France Van Wambeke[2], Moira Dion[3], and Solange Duhamel[1]

[1]Lamont-Doherty Earth Observatory, Division of Biology and Paleo Environment, Columbia University,
PO Box 1000, 61 Route 9W, Palisades, New York 10964, USA.
[2] Aix-Marseille Université, CNRS, Université de Toulon, IRD, OSU Pythéas, Mediterranean Institute of Oceanography (MIO),
UM 110, 13288, Marseille, France.
[3] Département de biochimie, de microbiologie et de bio-informatique, Faculté des sciences et de génie, Université Laval, 2325, rue
de l'Université, Québec (QC), G1V 0A6, Canada.

*Correspondence to*: Nicholas Bock (nab2161@ldeo.columbia.edu)

**Abstract:** Accounting for 40 percent of the earth's surface, oligotrophic regions play an important role in global biogeochemical cycles, with microbial communities in these areas representing an important term in global carbon budgets. While the general structure of microbial communities has been well documented in the global ocean, some remote regions such as the Western

Tropical South Pacific (WTSP), remain fundamentally unexplored. Moreover, the biotic and abiotic factors constraining microbial abundances and distribution remain not-well resolved. In this study, we quantified the spatial (vertical and horizontal) distribution of major microbial plankton groups along a transect through the WTSP during the austral summer of 2015, capturing important autotrophic and heterotrophic assemblages including cytometrically determined abundances of non-pigmented protists (also called flagellates). Using environmental parameters (e.g. nutrients and light availability) as well as statistical analyses, we

estimated the role of bottom-up and top-down controls in constraining the structure of the WTSP microbial communities in biogeochemically distinct regions. At the most general level, we found a "typical tropical structure," characterized by high abundances of Prochlorococcus at the surface, a clear deep chlorophyll maximum at all sampling sites, and a deep nitracline. Despite their relatively low abundances, picophytoeukaryotes (PPE) accounted for up to half of depth-integrated phytoplankton biomass in the lower euphotic zone. While present at all stations, Synechococcus accounted for only 2 % and 4 % of total phytoplankton

abundance and biomass, respectively. Our results show that the microbial community structure of the WTSP is typical of highly stratified regions, and underline the significant contribution to total biomass by PPE populations. Strong relationships between $N_2$ fixation rates and plankton abundances demonstrate the central role of $N_2$ fixation in regulating ecosystem processes in the WTSP, while comparative analyses of abundance data suggest microbial community structure to be increasingly regulated by bottom-up processes under nutrient limitation, possibly in response to shifts in abundances of high nucleic acid bacteria (HNA).


## 1 Introduction

Oligotrophic oceans, covering approximately 40 percent of the earth's surface, represent one of the earth's largest biomes. Despite relatively reduced productivity compared to coastal and upwelling regions, these nutrient-limited ecosystems account for an estimated 90 percent of global marine primary production (Karl et al., 2012). Therefore, it is of central importance to understand

the factors shaping microbial communities in these regions, and to account for how these factors might vary seasonally and geographically between the ocean's major oligotrophic regions. There has been an enormous deal of progress toward this goal over the last three decades. The groups of phytoplankton that numerically dominate the open ocean—*Prochlorococcus*, *Synechococcus*, and picophytoeukaryotes (PPE)— are of small size, generally <2 µm, with the most abundant group, *Prochlorococcus*, identified in the late 1980s (Chisholm et al., 1988). Since then, the widespread use of flow cytometry to characterize microbial communities has led to the publication of numerous studies documenting the distribution of these organisms (Veldhuis and Kraay,

2000). Especially over the last ten years, molecular methodologies have allowed for the characterization of these groups at the





taxon or species level, revealing enormous diversity across all trophic levels of marine microbial communities and identifying numerous ecotypes occupying distinct ecological niches (Carlson et al., 2007; Venter et al., 2004). More recently, environmental sequencing has revealed a surprising diversity of small sized picophytoeukaryotes (PPE) and new eukaryotic lineages continue to be discovered and characterized (Kashtan et al., 2014; Kim et al., 2016; Lepère et al., 2009; Rii et al., 2016b) while also reveal-

ing the importance of viruses in regulating phytoplankton communities (Brum et al., 2015; Huang et al., 2015).

However, the role of physical and biogeochemical processes in shaping microbial communities in the oligotrophic ocean remains unclear. While some general *in situ* trends are apparent— for example the predominance of *Synechococcus* and P PE in nutrient rich waters and the predominance of *Prochlorococcus* in nutrient depleted regions—spatial and temporal variability provide significant challenges to the generalization of these patterns at a global level (Fuhrman, 2009). And while there are

accounts of microbial community structure in oligotrophic regions of the North Atlantic (Partensky et al., 1996) and North Pacific (Campbell and Vaulot, 1993; Karl, 1999), as well as for the Mediterranean (Denis et al., 2010) and Arabian Seas (Campbell et al., 1998), the South Pacific remains less well documented. Although there are three reports on the distribution of bacteria, *Prochlorococcus*, *Synechococcus* and PPE abundance and biomass in the oligotrophic subtropical southeast Pacific (Grob et al., 2007a, 2007b; Lepère et al., 2009), and one paper describing cell abundance distribution of *Prochlorococcus*, *Synechococcus* and

PPE in the southwest Pacific near New Caledonia (Blanchot and Rodier, 1996), there are no reports on community structure in the oligotrophic to ultraoligotrophic regions of the Western Tropical South Pacific (WTSP). Moreover, despite their crucial role as grazers of microbial plankton, small-sized heterotrophic protists (i.e. non-pigmented cells between ~2 to 5 μm, hereafter HNF), also called flagellates, have received little attention (Christaki et al., 2011). Although there have been an increasing number of reports focusing on their role as predators over the last couple of decades, HNF are not routinely measured and their distri-

bution relative to other microbial groups is not well constrained (Christaki et al., 2011).

In this study, we present an account of microbial plankton community structure during late austral summer in distinct biogeochemical regions of the WTSP, ranging from mesotrophy to ultraoligotrophy. Our primary goal is to document the microbial community structure in the region, based on flow cytometry data capturing bacterioplankton (low-DNA-content, LNA, and high-DNA-content, HNA), phytoplankton (*Prochlorococcus*, *Synechococcus* and PPE) and HNF groups. In addition, we describe

the dominant biogeochemical gradients observed along the transect, and attempt to identify the physical and ecological variables influencing the abundance and distribution of plankton groups in the region. We also compare several previously published empirical models that make use of bacteria and HNF abundances to evaluate trophic interactions between populations of heterotrophic plankton groups.

**2. Material and Methods**

**2.1 Field sampling**

A zonal characterization of the biogeochemistry and biological diversity of the western tropical South Pacific (WTSP) was conducted along trophic gradients during the OUTPACE cruise (Oligotrophy to UlTra-oligotrophy PACific Experiment, DOI: http://dx.doi.org/10.17600/15000900, RV *L'Atalante*, February–April 2015) between New Caledonia and Tahiti (Moutin et al.,

2017). To describe the longitudinal and vertical distribution of different groups of pico- and nanoplankton, we sampled 15 short duration stations (SD1 to SD15, occupied during 8h, Fig 1). Three long duration stations (LDA, LDB, and LDC, occupied during 7 days), chosen for their contrasted biogeochemical conditions (Table 1), were sampled in Lagrangian mode (de Verneil et al., 2017a). All stations were sampled at 12 different depths from the surface down to 200 m for microbial characterization of the extended photic layer. The photic layer, $Z_e$ (m) is defined as the sunlit layer of the water column between the surface and the

depth where irradiance is reduced to 0.1 % of its surface value (hereafter $PAR_{0.1}$).





### 2.2 Pico- and nano- plankton analyses

For cell enumeration, duplicate 1.8-ml samples were fixed (0.25 % electron microscopy grade paraformaldehyde, w/v) for 10-15 min at room temperature and in the dark, flash-frozen in liquid nitrogen and stored at -80°C for later analysis using a BD Influx flow cytometer (BD Biosciences, San Jose, CA, USA). Pigmented groups, *Prochlorococcus*, *Synechococcus* and PPE, were enu-

merated in unstained samples for 5 min at ~61 μl/min. Bacteria were discriminated in a sample aliquot stained with SYBR Green I DNA dye (1:10,000 final) and enumerated for 1 min at ~65 μl/min. HNF were analyzed for 8 min at ~193 μl/min in a sample stained with SYBR Green I at 1:5,000 final concentration (Christaki et al., 2011; Zubkov et al., 2007). Particles were excited at 488 nm (plus 457 nm for unstained samples). Forward (< 15°) scatter (FSC), side (90°) scatter (SSC), green fluorescence (530/40 nm), orange fluorescence (580/30 nm) and red fluorescence (> 650 nm) emissions were measured. Pigmented groups were iden-

tified and enumerated based on their chlorophyll (red) fluorescence and FSC (size) signatures. The high phycoerythrin (orange) signal in *Synechococcus* was used to distinguish them from *Prochlorococcus* and PPE. Using a FSC detector with small particle option and focusing a 488 plus a 457 nm (200 and 300 mW solid state, respectively) laser into the same pinhole greatly improved the resolution of dim surface *Prochlorococcus* population from background noise (Duhamel et al., 2014). LNA and HNA bacteria were discriminated based on their low and high green fluorescence, respectively, in a SSC vs green fluorescence plot

(Vazquez-Dominguez et al., 1999; Van Wambeke et al., 2011). In samples from the upper euphotic layer, where *Prochlorococcus* signal overlapped with HNA in a SSC or FSC vs green fluorescence plot, *Prochlorococcus* abundance counted in unstained samples were subtracted from the HNA abundance enumerated in a larger gate. Total bacteria refers to the sum of LNA and HNA abundances. Reference beads (Fluoresbrite, YG, 1 μm) were added to each sample and red fluorescence from chlorophyll and FSC values are presented relative to the reference beads (arbitrary units, A.U.).

$N_2$ fixation rates were measured in triplicate at all stations (except SD13) using the $^{15}N_2$ isotopic tracer technique (adapted from Montoya et al., 1996). Briefly, seawater samples were collected in HCl-washed, sample-rinsed (3 times) light-transparent polycarbonate 2.3-L bottles from 6 depths (75 %, 50 %, 20 %, 10 %, 1 %, and 0.1% surface irradiance levels), sealed with caps fitted with silicon septa and amended with 2 mL of 98.9 atom % $^{15}N_2$ (Cambridge isotopes). Incubation bottles were incubated in on-deck incubators equipped with circulating seawater at the specified irradiances using blue screening. Incubations

were stopped by filtration of the entire sample onto precombusted 25-mm GF/F glass fiber (Whatman, 0.7 μm nominal pore size) filters, which were then analyzed for $^{15}N/^{14}N$ ratios and PON concentrations using an elemental analyzer coupled to a mass spectrometer (EA-IRMS, Integra CN, SerCon Ltd) as described in Bonnet et al. (this issue).

### *2.3 Data analyses and statistics*

All data analyses were performed using R Studio (R Core Team, 2013). All figures were prepared using the ggplot2 package (Wickham, 2009), with the exception of the contour plots presented in figure 2, which were prepared using Ocean Data View 4.7.10 (Schlitzer, 2017). Following the designation used in Grob et al. (2007b), stations along the transect were characterized as either mesotrophic (> 0.1 mg Chl *a* m$^{-3}$) or oligotrophic (< 0.1 mg m$^{-3}$). Based on MODIS average surface chlorophyll a (Chl *a*) for March 2015, stations 1 through 12, along with long duration station A, were categorized as mesotrophic stations (mean Chl *a*

0.104 ± 0.014 mg m$^{-3}$), while stations 13 through 15, along with long duration station C, were categorized as oligotrophic stations (mean Chl *a* 0.028 ± 0.002 mg m$^{-3}$). Long duration station B was in the late stages of a phytoplankton bloom at the time of sampling (de Verneil et al., 2017b), and consequently featured significantly higher Chl *a* concentrations than other stations on the transect (0.197 ± 0.001 mg m$^{-3}$). As such, it was analyzed independently of other stations on the transect, and is simply referred to as LDB.





Chlorophyll fluorescence of microbial groups were calculated as the ratio of mean red fluorescence of each cell population to that of the reference beads, based on flow cytometry results. Per-cell biomass was calculated using previously published conversion factors: 29 fg C per *Prochlorococcus* cell, 100 fg C per *Synechococcus* cell, 11.5 fg C per bacterial cell, and 1500 fg C per PPE cell (Zubkov et al., 2000). Biomass contributions of different phytoplankton groups were estimated by multiplying

cell abundances by the conversion factors above and integrating by depth across the euphotic zone. To account for larger eukaryotes not captured by cytometry, we used the method described in Vidussi et al (2001) to estimate the relative biomass contributions of diatoms and dinoflagellates based on concentrations of fucoxanthin and peridinin relative to those of total diagnostic pigments: zeaxanthin*, dv chl *a*, Tchl *b*, 19' hexanoyloxyfucoxanthin (19'-HF), 19' butanoyloxyfucoxanthin (19'-BF), alloxanthin, fucoxanthin, and peridinin. Pigment concentrations were measured via high performance liquid chromatography, as described

elsewhere (Ras et al., 2008).

Concentrations of $PO_4$, $NO_2$, and $NO_3$ were measured using an SEAL AA3 HR auto-analyzer (SEAL Analytical, UK), as described by Moutin et al. (this issue). Abundances, biomass, and nutrient values are reported either as depth-integrated totals or as depth-normalized averages. In order to account for large vertical gradients in both abundances and nutrient concentrations within the $Z_e$, integrations were performed across two depth ranges: the upper photic zone ($Z_u$), integrating from the surface to

the recorded 2.7 % isolume (hereafter $PAR_{2.7}$) and the lower euphotic zone ($Z_l$), integrating from $PAR_{2.7}$ to the recorded 0.1 % isolume (hereafter $PAR_{0.1}$).The mixed layer depth was measured continuously along the transect, as described by de Verneil, et al. (2017a). The mixed layer was entirely within $Z_u$ at all stations (Fig. 2), while the nitracline, defined as the depth where measurements of $NO_2 + NO_3$ (hereafter $NO_x$) first exceed 0.1 μM, occurred within $Z_l$ at all stations (Fig. 2F). Attenuation coefficients, $k$, were calculated by using CTD PAR measurements to solve the Beer-Lambert equation between surface PAR and that corre-

sponding to $PAR_{0.1}$.

To maximize the power of statistical tests, depth-integrated values were calculated for individual casts, without averaging results from long duration stations. Integration results were then assigned categorical variables corresponding to biogeochemical condition or euphotic zone layer, with two-way ANOVA being used to identify statistical relationships between parameters according to these variables. To ensure that data met the requirements for ANOVA (i.e. normally distributed, and with equal variance between groups), values were log transformed before performing statistical tests. In the case of nutrient data and $N_2$ fixation rates, which were often skewed by large occurrences of small values, data were transformed using the formula data = Log (data*100 + 1). Bivariate comparisons between biogeochemical parameters were performed using Pearson's correlation. The Shapiro-Wilk test was used to assess normality, while the Levene's test was used to confirm homogeneity of variance.

## 3. Results

### 3.1 Physico-chemical characteristics of the studied area

Stations along the transect were characterized by warm sea surface temperatures (mean 29.4 ± 0.4 °C) and strong water column stratification. Stations between 160° E and 170° W (stations SD1 to SD12, LDA, and LDB) featured a prominent thermocline at approximately 25 m, which increased in depth to approximately 50 m at the oligotrophic stations (LDC, and SD13 to SD15).

This increase was mirrored in the mean mixed layer depth, which ranged from 21 ± 5 m at mesotrophic sites to 25 ± 8 m at oligotrophic sites (Fig. 2). There was also a significant west to east decrease in light attenuation ($k$), ranging from 0.059 ± 0.006 at the mesotrophic stations to 0.044 ± 0.005 at the oligotrophic stations. An exception to this trend was found at LDB, where $k$ increased to 0.078 ± 0.021. Corresponding to these changes in $k$, $PAR_{0.1}$ deepened from west to east, ranging from 113 ± 13 m at mesotrophic stations to 178 ± 5 m at oligotrophic sites. Again, LDB presented an exception to this general trend, where $PAR_{0.1}$

was recorded at 83 ± 5 m.



All stations across the transect featured a prominent deep chlorophyll maximum (DCM). Mirroring changes in $k$, the DCM showed a general increase in depth from west to east, ranging from $85 \pm 20$ m to $133 \pm 20$ m from the mesotrophic to the oligotrophic stations, respectively. The DCM depth at LDB was also an exception, decreasing to $50 \pm 19$ m. The concentration of dissolved oxygen was near equilibrium with the atmosphere near the surface, becoming slightly oversaturated below the mixed

layer. This subsurface maximum occurred at a mean depth of $55 \pm 18$ m and was weakly correlated with the depth of the DCM (Pearson's r = 0.44, p < 0.01). Oxygen levels within $Z_e$ were above 158.31 mol kg$^{-1}$ across the entire transect, with there being no suboxic regions at any of the stations sampled. The nitracline generally tracked the DCM, occurring at depths ranging from $93 \pm 17$ m at mesotrophic stations to $127 \pm 13$ m at oligotrophic stations. The nitracline was decoupled from the DCM at LDB, where it occurred at $108 \pm 22$ m.

Nitrate plus nitrite ($NO_X$) concentrations were depleted in $Z_u$ across all biogeochemical conditions. While depth-averaged $NO_X$ were significantly elevated in $Z_l$ at mesotrophic and oligotrophic stations, (ANOVA, p < 0.01), no significant difference was encountered between $Z_u$ and $Z_l$ at LDB (Table 1). Depth averaged phosphate ($PO_4$) concentrations in $Z_u$, were significantly elevated at oligotrophic sites relative to other stations (ANOVA, p < 0.01), although no significant differences were identified between mesotrophic stations and LDB.

**3.2 Phytoplankton community structure**

*Prochlorococcus* dominated phytoplankton abundances at all sampling sites, with average $Z_e$-integrated abundances being two orders of magnitude greater than those of *Synechococcus* and PPE (Table 1). $Z_e$-integrated *Prochlorococcus* abundances ranged from 130 x10$^{11}$ cells m$^{-2}$ at SD3 to 270 x10$^{11}$ cells m$^{-2}$ at SD1, while those for *Synechococcus* ranged from 0.65 x10$^{11}$ cells m$^{-2}$ at SD15 to 18.45 x10$^{11}$ cells m$^{-2}$ at LDB. PPE abundances ranged from 1.21 x10$^{11}$ to 2.35 x10$^{11}$ cells m$^{-2}$ at SD3

and SD12, respectively. There were no significant differences in $Z_e$-integrated abundances of these groups across biogeochemical conditions, except for those of *Synechococcus*, which were significantly greater at LDB compared to mesotrophic or oligotrophic stations (ANOVA, p < 0.01). Transect-wide, *Prochlorococcus* accounted for approximately $97 \pm 2$ % of total phytoplankton cells enumerated by flow cytometry. *Synechococcus* and PPE accounted for $2 \pm 2$ % and $0.8 \pm 0.2$ % of total phytoplankton abundance, respectively. Pooling all data, statistically significant correlations were identified between all pairs of plankton

ton groups (Pearson's R, p < 0.01).

Based on $Z_e$-integrated abundances, relative contributions of different phytoplankton groups to total phytoplankton abundance showed considerable longitudinal variation. *Synechococcus* accounted for $0.4 \pm 0.1$ % of phytoplankton cells at oligotrophic stations, $2.4 \pm 1.9$ % at mesotrophic stations, and $7.4 \pm 4.4$ % at LDB. *Prochlorococcus* abundances, by contrast, represented $92.1 \pm 4.3$ % of phytoplankton cells at LDB and, $96.7 \pm 2.0$ % at mesotrophic stations, and $98.6 \pm 0.2$ % at oligotrophic

stations. Ratios of $Z_e$-integrated abundances of *Prochlorococcus* to *Synechococcus* varied significantly (ANOVA, p < 0.01) between oligotrophic stations ($235.7 \pm 65.1$) and LDB ($16.2 \pm 10$). PPE abundances showed less variability, with relative abundances ranging from $0.4 \pm 0.1$ % to $0.9 \pm 0.1$ % of phytoplankton cells at LDB and oligotrophic stations, respectively. Differences in the relative abundance of PPE between biogeochemical conditions were not statistically significant. Statistically significant negative correlations were found between concentrations of $NO_X$ and $PO_4$ and all plankton groups (Pearson's R, p < 0.01), while

significant positive correlations were identified between $N_2$ fixation rates and abundances of *Prochlorococcus* and heterotrophic bacteria (Pearsons' R, p < 0.01). These correlations persisted when subsetting data to include mixed layer values alone (Pearson's R, p < 0.01), with the exception of correlations between $NO_x$ and plankton groups.

Most stations were characterized by a well-defined two-tier distribution of phytoplankton within the $Z_e$ (Fig. 3), with depth-integrated abundances of *Prochlorococcus* and *Synechococcus* being greatest in the $Z_u$, and PPE abundances being greatest

in $Z_l$ (Table 1). These differences between $Z_u$ and $Z_l$ abundances were found to be statistically significant for *Prochlorococcus*





across all conditions. Differences were significant for *Synechococcus* at mesotrophic stations, and for PPE at oligotrophic and mesotrophic stations (ANOVA, p < 0.01). *Prochlorococcus* and PPE abundances showed subsurface maxima at both mesotrophic and oligotrophic stations. Averaging across the transect, *Prochlorococcus* abundance maxima occurred at depths corresponding to 24.2 ± 24.4 % PAR while PPE maxima occurred at depths corresponding to 0.6 ± 0.4 % PAR. Depths of these max-

ima showed a west to east increase, and were significantly deeper at oligotrophic stations than at mesotrophic stations for all phytoplankton groups (ANOVA, p < 0.01).

There was no significant variation in $Z_e$ integrated biomass between different conditions, although $Z_u$-integrated biomass was significantly greater (p < 0.01) at LDB compared to mesotrophic and oligotrophic stations (Fig. 4). In keeping with relative abundances, *Prochlorococcus* cells represented the greatest fraction of $Z_e$ biomass, accounting for an average of 77.1 ±

5.5 % across the transect. By comparison, PPE accounted for an average of 18.7 ± 5.4 % of $Z_e$ biomass, while *Synechococcus* accounted for 3.9 ± 4.3 %. However, there was considerable vertical and longitudinal variation in these trends, especially in contributions to total biomass by PPE and *Synechococcus* populations (Fig. 4). PPE accounted for 29 ± 14 % of phytoplankton biomass considering $Z_l$ alone, and up to 64 % of $Z_l$ biomass at SD4. *Synechococcus* accounted for up to 13 % of $Z_u$ phytoplankton biomass at LDB.

**3.3 Distributions of bacterioplankton and HNF**

$Z_e$-integrated bacterial abundances ranged from 271 x$10^{11}$ to 585 x$10^{11}$ cells m$^{-2}$ at SD4 and LDB, respectively (Table 1). Despite this range, there was relatively little variation when comparing biogeochemical regions; while average $Z_e$-integrated abundances at oligotrophic stations were somewhat elevated compared to those for mesotrophic stations, and while those at LDB were amongst highest on the transect, these differences were not statistically significant. Examining HNA and LNA subpopulations,

$Z_e$ integrated abundances for HNA bacteria ranged from 80 x$10^{11}$ to 291 x$10^{11}$ cells m$^{-2}$ at SD4 and SD9, respectively, while values for LNA ranged from 150 x$10^{11}$ cells m$^{-2}$ at LDA to 298 x$10^{11}$ cells m$^{-2}$ at SD8. As with total bacteria, there were no statistically significant longitudinal differences in $Z_e$ integrated HNA or LNA abundances when comparing different biogeochemical regions. However, the fraction of HNA to total bacteria (%HNA) showed considerable longitudinal variation, ranging from 41.1 ± 2.1 % to 48.0 ± 4.9 % between oligotrophic stations and LDB, respectively (Fig. 5). Values for %HNA were significantly

greater at LDB relative to the mesotrophic and oligotrophic stations (ANOVA, p < 0.01) and at mesotrophic stations relative to oligotrophic stations (ANOVA, p < 0.01). Bacterial abundances showed less variability with depth than did phytoplankton groups, with there being no significant differences in depth-integrated abundances between $Z_u$ and $Z_l$ for total bacteria, HNA, or LNA. Depth profiles of %HNA, however, were more variable than those for total bacteria. %HNA increased from the surface to PAR$_{0.1}$ across all biogeochemical regions, and distinct local minima were apparent near the DCM at oligotrophic and meso-

trophic stations.

Mean HNF abundances in $Z_u$ ranged from 0.38 x $10^3$ to 2.3 x $10^3$ cells ml$^{-1}$ at SD 15 and LDB, respectively. $Z_e$ integrated abundances at LDB were significantly greater than those at oligotrophic stations (ANOVA, p < 0.01), although no significant difference was found between oligotrophic stations or LDB and the mesotrophic stations. Depth integrated abundances were greater in the $Z_u$ than the $Z_l$ at mesotrophic and oligotrophic stations (t-test, p < 0.01), but there was no significant difference at

LDB.

**3.4 Bottom-up vs. top-down control of microbial communities**

In order to assess the roles of top-down and bottom-up control over microbial group abundances along the transect, we used a combination of approaches based on previously published models. The model described by Gasol (1994) was used to assess top-

down vs bottom-up control of HNF abundance (Fig. 6). Specifically, this approach compares observed ratios of bacteria to HNF



with theoretical maxima determined by a Lotka-Volterra model. The main assumption of the model is that bacteria to HNF ratios nearer to theoretical maxima implies increased bottom-up control of HNF by bacterial abundance. This difference is quantified with the parameter $d$, which is calculated as the difference between theoretical and observed HNF abundances. Small values of $d$ are thus interpreted as being indicative of top-down control on bacterial populations by HNF, or by a significant use of resource other than bacteria by HNF. Large values of $d$ are interpreted as being indicative of a decoupling between the two groups, and/or a top down control of HNF by their predators, like ciliates. Average $Z_u$ values for $d$ were $0.59 \pm 0.11$, $0.62 \pm 0.19$, and $0.80 \pm 0.09$ for mesotrophic, bloom, and oligotrophic stations respectively. By contrast, average $Z_l$ values for $d$ were $0.46 \pm 0.14$, $0.43 \pm 0.08$, and $0.45 \pm 0.10$ for mesotrophic, bloom, and oligotrophic stations respectively. $Z_u$ Values for $d$ were significantly elevated at the oligotrophic stations relative to mesotrophic stations and LDB (ANOVA, $p < 0.01$). No significant differences in $d$ were identified between biogeochemical regions in $Z_l$.

Regressions between abundances of bacteria and HNF were measured for $Z_u$ and $Z_l$ across biogeochemical conditions (data not shown). The variability in HNF abundances explained by bacteria abundance was generally greater in $Z_l$ compared to $Z_u$. $Z_u$ bacterial abundances explained 24 %, 30 % and 30 % of variability in HNF abundance at mesotrophic, LDB, and oligotrophic stations, respectively. In $Z_l$, bacterial abundances explained 57 % and 72 % of variability at mesotrophic and oligotrophic stations, respectively, while this relationship was not statistically significant at LDB. Repeating this procedure for HNA bacteria alone, $Z_u$ HNA abundances were found to explain 15 %, 29 %, and 73 % of variability in HNF abundance at mesotrophic, LDB, and oligotrophic stations, respectively. $Z_l$ bacteria abundances were found to explain 61 % of variability in HNF populations at oligotrophic stations, while relationships at mesotrophic stations and LDB were insignificant. $Z_u$ values for %HNA explained 73 % of variability in HNF abundances at oligotrophic stations, while this relationship was weak and insignificant at mesotrophic stations and LDB.

Using the ciliate abundances collected by Dolan et al. (2016) during the OUTPACE cruise, ratios of depth-integrated abundances of ciliates to protists (with protists abundances multiplied by $10^{-11}$ for readability) were found to range from 2.8 in the upper euphotic zone at LDB to 17.6 in the lower euphotic zone at LDC. In the upper euphotic zone, this ratio increased from 2.9 at LDB to 10.0 at LDC and 10.9 at LDA. The lower euphotic zone showed a slightly different pattern, with the ratio increasing from 9.0 at LDA to 9.6 at LDB and 17.5 at LDC. Because data available were limited to one set of measurements at each of those three stations, it was not possible to determine whether differences in these results were statistically significant. However, comparing differences between biogeochemical conditions based on observations at individual depths, rather than depth-integrated values, indicated the ratio of ciliates to protists to be significantly lower at LDB compared to LDC (ANOVA, $p < 0.01$). Differences between other biogeochemical conditions, however, were statistically insignificant. No significant vertical or longitudinal differences were identified for ratios of protists to cyanobacteria, nor for ratios of bacteria to cyanobacteria.

### 3.5 Distribution of pigments and photo acclimation in different phytoplankton groups

Phytoplankton group-specific relative fluorescence values obtained by flow cytometry for *Prochlorococcus*, *Synechococcus*, and PPE, showed significant (t-test, $p < 0.01$) increases with depth across all biogeochemical conditions (Fig. 3). Phytoplankton relative fluorescence for all groups showed little variation within the $Z_u$, although marked increases occurred in the region of $PAR_{2.7}$. *Prochlorococcus* relative fluorescence showed a continuous increase to 200 m at mesotrophic and oligotrophic stations, and an increase to 150 m at LDB. *Synechococcus* and PPE showed clear maxima near or just below $PAR_{0.1}$, although in oligotrophic samples PPE relative fluorescence showed a continuous increase to 200 m.



Analysis of HPLC pigments data using the approach described in Vidussi et al. (2001) largely mirrored our flow cytometry results. Transect-wide, zeaxanthin and chlorophyll *b*, pigments corresponding to cyanobacteria and prochlorophytes dominated in $Z_u$, accounted for 80 ± 5.1 % of total diagnostic pigments. Fucoxanthin and peridinin, diagnostic of diatoms and dinoflagellates, accounted for 3.8 ± 1.0 %. Concentrations of 19'HF and 19'BF—diagnostic pigments typically used to assess abundances of prymnesiophytes and chrysophytes/pelagophytes, respectively (Wright and Jeffrey, 2006)—showed significant horizontal and vertical variability (Table 2). Absolute concentrations of both pigments showed significant increases with depth at mesotrophic and oligotrophic stations (ANOVA, p < 0.01), although increases at LDB were not statistically significant. Ratios of 19'HF:Chl *a* were significantly greater than those of 19'BF:Chl *a* across all biogeochemical conditions (t-test, p < 0.01). Ratios of 19'HF:Chl *a* showed significant increases with depth at mesotrophic stations (ANOVA, p < 0.01), although increases at oligotrophic stations and LDB were not statistically significant. Ratios of 19'BF:TChl *a* showed significant increases at mesotrophic and oligotrophic stations (ANOVA, p < 0.01), while increases at LDB were insignificant. $Z_l$ ratios of 19'HF:19'BF were significantly elevated at LDB compared to mesotrophic stations (ANOVA, p < 0.01). That no similar such difference was observed in comparing LDB to oligotrophic stations is likely the result of the reduced number of samples available for making this comparison. Indeed, the difference was nearly significant (ANOVA, p = 0.03), while $Z_l$ ratios of 19'HF:19'BF at mesotrophic stations were remarkably similar (ANOVA, p = 0.99). A moderately strong relationship was identified between carotenoid concentrations and PPE abundances, with variability in PPE accounting for 46 % of variability in 19'BF + 19'HF (p < 0.01).

## 4 Discussion

### 4.1 Distribution of phytoplankton populations in the WTSP and relative contribution to biomass

The WTSP is representative of permanently stratified conditions (Sarmiento et al., 2004), implying slow nutrient delivery to the euphotic zone and rapid nutrient recycling within the mixed layer (Moutin et al., this issue). Accordingly, transect-wide biogeochemical conditions captured by our data were similar to those of the "typical tropical structure" described by Herbland and Voituriez (1979), featuring large abundances of pico-sized organisms in $Z_u$, a deep nitracline, and a prominent DCM. Differences in relative abundances of phytoplankton groups between $Z_u$ and $Z_l$ showed a clear two-tiered vertical niche partition, with *Prochlorococcus* and *Synechococcus* reaching maximum abundances in the $Z_u$ and PPE achieving maximum abundances in the $Z_l$. This vertical distribution has been well documented in other regions, and is thought to be characteristic of highly stratified oligotrophic systems (Dore et al., 2008; Painter et al., 2014; Partensky et al., 1996). Based on estimates from HPLC data, larger organisms such as diatoms and dinoflagellates, were present in very low abundance along the transect, in comparison to small-sized phytoplankton.

Although the use of different conversion factors for estimating per-cell carbon makes it difficult to compare between different studies, our biomass estimates largely agree with those reported for other oligotrophic regions (Grob et al., 2007b; Partensky et al., 1996; Pérez et al., 2006; Zubkov et al., 2000). *Prochlorococcus* accounted for the large majority of phytoplankton biomass in the $Z_u$, with *Synechococcus* and PPE only making relatively minor contributions. In $Z_l$, by contrast, PPE provided a more sizeable and occasionally dominant share to phytoplankton biomass. This effect was particularly pronounced at oligotrophic stations, where increases in $Z_l$ PPE biomass compensated for reductions in $Z_u$ *Prochlorococcus* biomass, resulting in $Z_e$ biomass totals to be similar to those for mesotrophic stations.

To compare our results to those of other regions of the global ocean, we conducted a literature search to identify other papers reporting $Z_e$-integrated abundances of *Prochlorococcus*, *Synechococcus*, and PPE. Based PPE abundances were similar to those reported under similar environmental conditions, perhaps as a result of low abundances in $Z_u$. The high abundances of *Pro-*



*chlorococcus* measured along the transect were generally found to be typical of oligotrophic regions, as were the observed abundances of *Prochlorococcus* relative to other phytoplankton groups (Table 3). However, the relative abundances of *Prochlorococcus* encountered at LDB were considerably higher than might be expected based on relative abundances reported for regions with comparable Chl *a* concentrations. This may be due to the transient nature of the bloom observed at LDB, which may make the

site incomparable to regions with more persistent inputs of nutrients. Alternatively, it may be the result of having captured the bloom in decay (de Verneil et al., 2017b) , with nutrients having been largely depleted and relative *Prochlorococcus* abundances returning to levels more representative of the WTSP region. Some variation between studies may also result from the instrumentation used, with earlier cytometers generally being thought to underestimate weakly fluorescent *Prochlorococcus* cells near the surface. Regardless, with $Z_e$ integrated *Prochlorococcus* abundances at LDB being greater than any others encountered in the

literature, these results highlight the importance of transient, localized blooms to cyanobacterial abundance in the region. The observed increase in ratios of $Z_e$-integrated *Prochlorococcus* abundances relative to those of *Synechococcus* from LDB to oligotrophic stations captures a global trend. The proportion of *Prochlorococcus* cells accounting for total phytoplankton abundances have generally been reported to decrease with increased Chl *a* concentrations, while those of *Synechococcus* cells have been found to increase along the same gradient (Table 3). In the Sargasso Sea, where winter mixing allows for the resupply of surface

nutrients, long-term studies have captured this relationship as a seasonal pattern, with ratios of *Prochlorococcus* to *Synechococcus* increasing inversely with changes in the depth of the nitracline (Campbell et al., 1998; Durand et al., 2001). The same phenomenon has been reported along biogeochemical conditions in the North Atlantic (Partensky et al., 1996; Zubkov et al., 2000), as well as the South East Pacific (Grob et al., 2007a; Rii et al., 2016b). By contrast, sites with comparatively limited seasonal variability, like Station ALOHA in the North Pacific subtropical gyre, have shown consistently high ratios of *Prochlorococcus* to

*Synechococcus* ratios year-round (Rii et al., 2016a), while studies of nitrate-rich eutrophic regions often report the complete exclusion of *Prochlorococcus* by *Synechococcus* and eukaryotic populations (Sherr et al., 2005; Zubkov et al., 2007).

**4.2 Potential factors regulating the horizontal distribution of phytoplankton groups**

In examining the potential factors regulating the distribution and abundance of cyanobacterial groups, our data did not reward

any expectations that abundances of *Prochlorococcus*, *Synechococcus* or PPE would correlate meaningfully with $NO_x$ concentrations. While negative relationships were identified between plankton abundances and $NO_x$ concentrations across all sites, this was likely the result of changes in these parameters with depth rather than being indicative of any causal relationship. Indeed, comparing values in $Z_u$ alone, correlations between $NO_x$ and plankton abundances became insignificant, with $NO_x$ being largely depleted above $PAR_{2.7}$. However, correlations between $N_2$ fixation, $PO_4$ concentrations, and plankton abundances persisted even

when subsetting data to only include measurements within the mixed layer, indicating covariation between these parameters to occur independently of depth.

Specifically, correlations between $N_2$ fixation rates and abundances of cyanobacteria suggest plankton abundances in the surface to respond to diazotroph-derived nitrogen (ammonia and DON) provided by $N_2$-fixing organisms, notably *Trichodesmium* which dominated in the mesotrophic $Z_u$ (Stenegren et al., 2017). Previous studies have demonstrated growth to increase

with DON enrichment in both *Synechococcus* and *Prochlorococcus* cultures (Moore et al., 2002), while others have indicated that diazotrophs may provide a large enough input of fixed nitrogen to sustain large populations of cyanobacteria (Bonnet et al., 2016b). Moreover, previous experiments in the New Caledonia lagoon showed a rapid transfer (24-48h) of recently fixed N by *Trichodesmium* towards non diazotrophic phytoplankton and heterotrophic bacteria (Bonnet et al., 2016a). Biological nitrogen inputs may allow for a more complete utilization of $PO_4$ at sites featuring high nitrogen-fixation rates (Mather et al., 2008;





Moutin et al., this issue) , accounting for the negative correlations observed between $PO_4$ concentrations and abundances of *Prochlorococcus* and *Synechococcus,* as well as for the negative correlations observed between $PO_4$ concentrations and $N_2$ fixation rates. These results, along with the low DIP turnover rates reported, suggest intense competition for phosphorus within the mixed layer, and a rapid transfer of fixed N toward heterotrophic bacteria (Van Wambeke et al, this issue).

### 4.3 Potential factors regulating vertical variability in phytoplankton community structure

Considerable variation in vertical distributions of phytoplankton groups was observed between biogeochemical regions. Although *Synechococcus* and PPE appeared confined to high-light and low-light depths, respectively, *Prochlorococcus* abundances showed a greater deal of vertical variability, with *Prochlorococcus* subsurface abundance maxima varying widely with respect to

PAR (Fig. 3). This indicates *Prochlorococcus* distributions in $Z_u$ to be less sensitive to changes in light availability than other phytoplankton groups (Partensky et al., 1999), possibly as a result of comparatively reduced increases in per-cell chlorophyll concentrations with depth, as reflected by relative fluorescence values (Fig. 3). However, the observed *Prochlorococcus* distributions reflects the average distribution of a mosaic of different ecotypes encompassing high diversity in their response to nutrients, light, and temperature (Johnson, 2006; Kashtan et al., 2014; Moore et al., 2002). Indeed, the increase in depth of *Prochlorococ-*

*cus* abundance maxima observed at oligotrophic stations is likely the result of the deepening of the euphotic layer, combined with the reduction of high-light ecotypes in $Z_u$. While previous studies have reported correlations between *Prochlorococcus* abundance maxima and nitracline depth (Li, 1995; Olson et al., 1990), no similar such patterns were observed in our data. These distributions may be a transient feature formed during restratification following winter mixing (Partensky et al., 1999), and are unlikely to be in response to nitrate availability, given the small nitrate utilization by *Prochlorococcus* in natural samples (Casey

et al., 2007). These results may also be the consequence of the difficulty in detecting weakly fluorescent high-light *Prochlorococcus* following earlier flow cytometry protocols with less sensitive instruments.

In contrast to other phytoplankton groups, PPE abundances were marginal in $Z_u$, but increased dramatically below $PAR_{10}$, reaching maximal abundances at depths closely correlated with those of the DCM. The lack of variability of PPE abundance maxima relative to PAR, along with the decoupling of PPE maxima from the nitracline at LDB, suggest PPE abundances

to be primarily controlled by light levels rather than by the availability of dissolved nutrients. However, it is difficult to consider these factors independently with the increased chlorophyll concentrations required at low light levels likely increasing nitrogen requirements on shade-adapted organisms (Edwards et al., 2015).

Differences in vertical distributions of PPE between stations likely also reflects variability in the composition of PPE communities. The decrease with depth observed in ratios of 19'HF:19'BF would suggest prymnesiophytes to dominate in the $Z_u$,

with chrysophytes and pelagophytes accounting for a greater proportion of total PPE abundance in $Z_l$. Similar distributions have been reported elsewhere, and have been suggested to reflect control of chrysophyte and pelagophyte abundances by nitrate availability (Barlow et al., 1997; Claustre et al., 1994; Marty et al., 2002). This interpretation agrees with our results, where the separation of PPE abundance maxima from the nitracline coincided with significantly elevated $Z_l$ ratios of 19'HF:19'BF compared to mesotrophic stations. That elevated $Z_l$ values for 19'HF:19'BF at LDB coincide with transect-wide maximal concentrations of

$NH_4$ (Moutin et al., this issue) suggest that prymnesiophytes may preferentially utilize reduced forms of nitrogen. This would also account for the elevated abundances of this group in $Z_u$ across all biogeochemical conditions, where reduced forms of nitrogen would generally be expected to be more abundant as a result of nutrient recycling. Admittedly, with variability in 19'HF+19'BF only accounting for ~50 percent of variability in picoeukaryote abundances, it is likely that observed patterns of 19'HF:19'BF capture changes in nano- and micro-sized eukaryotes in addition to PPE.

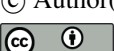



It is also possible that $Z_l$ PPE populations are responding to the availability of nitrogen fixed by UCYN-A cyanobacteria, which were reported to have distributions at least qualitatively similar to those of PPE across the transect (Stenegren et al., 2017). Several UCYN-A clades have been identified to form symbioses with small prymnesiophytes, including at least one pico-sized haptophyte (Martínez-Pérez et al., 2016), making it seem plausible that such relationships could play an important role in

controlling PPE distributions.

**4.4 Factors controlling bacterial abundance and the role of heterotrophic protists**

Average bacterioplankton abundances in the $Z_e$ ($3.6 \times 10^8 \pm 2.6 \times 10^5$ cells $l^{-1}$) were within the established range of $1\text{-}5 \times 10^8$ cells $l^{-1}$ for the oligotrophic ocean (Ducklow, 2002). That bacterioplankton abundances at LDB should exceed this range, slightly exceeding $7.0 \times 10^8$ cells $l^{-1}$, is not surprising, based on the increased abundances of phytoplankton at that station. Surface values

for %HNA, ranging from 30 % at SD15 to 51 % at SD1, are similar to previously reported values, as is the observed increase in %HNA with depth (Van Wambeke et al., 2011). The reduced values for %HNA reported for oligotrophic stations are similar to those reported for other nutrient-limited regions, and may be the result of LNA cells being capable of more rapid growth than HNA under nutrient limitation (Andrade et al., 2007; Nishimura et al., 2005).

To assess variation in trophic interactions between HNF and bacteria across biogeochemical conditions, we used the

method presented by Gasol (1994), which compares observed ratios of HNF to bacteria with theoretical maxima (*d*). Previous applications of this model have demonstrated an increase in top-down control of bacterial populations in low-chlorophyll regions, demonstrated by low values for *d* under nutrient limitation (Gasol et al., 2002). These results find general support in the literature, on the underlying assumption that nutrient-limited regions are characterized by reduced abundances of top predators, resulting in increased grazing pressure on bacteria via a trophic cascade (Pernthaler, 2005). Our data, however, contrast with

these conclusions, with values for *d* being significantly greater at oligotrophic stations relative to those corresponding to mesotrophic stations or LDB, suggesting a reduction in grazing pressure with increased nutrient limitation in $Z_u$.

Based on the significantly reduced $Z_u$ values for %HNA encountered at the oligotrophic stations, decoupling of bacterial and HNF populations may be the result of diminished prey quality at these sites. While it has been debated whether HNA and LNA can be interpreted as representing active and inactive cells, respectively (Jochem et al., 2004; Vaqué et al., 2001), HNA

bacteria have generally been found to be larger in diameter than LNA bacteria (Van Wambeke et al., 2011), possibly making them more susceptible to grazing. Such a phenomenon has been described previously in nutrient-limited regions (Longnecker et al., 2010; Vaqué et al., 2001), although studies conducted in relatively nutrient-rich regions have reported no similar such evidence (Jochem et al., 2004). In our data, the increased $r^2$ values for linear regressions between abundances of HNA and protists at oligotrophic stations suggest that HNA densities may be more important in controlling HNF densities at oligotrophic sites than at

mesotrophic stations or LDB. This would seem to suggest an increase in the importance of bottom-up processes in regulating plankton abundances under nutrient limitation. This would also account for the increased values of *d* calculated at oligotrophic stations—the Lotka-Volterra equation used to establish the model's theoretical maxima—not accounting for selective grazing on subpopulations of bacteria under different trophic conditions, and thereby providing an overestimate of the abundance of HNF to be expected for the observed abundance of total bacteria.

By reducing bacterial abundances relative to those of HNF, the reduction of *d* reported in $Z_l$ may also result from phagotrophy by PPE. Feeding experiments in the North Atlantic have demonstrated small plastidic eukaryotes to account for up to 90 % of bacterivory in nutrient-limited waters (Zubkov and Tarran, 2008), while laboratory and field studies have demonstrated increased feeding rates specifically in response to P limitation (Christaki et al., 1999; McKie-Krisberg et al., 2015). While PPE abundance maxima always occurred below the phosphacline, it seems probable that N limitation could result in a similar re-



sponse. Such a phenomenon is admittedly difficult to demonstrate with our data, though, as we did not perform grazing experiments. While ratios of bacteria to PPE show dramatic minima in the $Z_l$ across all stations, there are no significant differences in these ratios that might suggest such a feeding response. That said, with bacterial abundances being two orders of magnitude greater than those of PPE, it may be unlikely that such a relationship could be inferred from abundance ratios alone.

We also cannot exclude that the decoupling of bacterial and HNF populations may reflect increased grazing pressure on HNF by ciliates, which would imply an increase in the importance of top-down processes under nutrient limitation. It is impossible to eliminate this possibility based on the model alone. However, given that ratios of ciliates to bacteria are similar between mesotrophic and oligotrophic stations, it does not seem likely that the significant differences in $d$ between these sites reflect a change in the interactions between these organisms. That said, it may also be the case that these differences result from changes

in the availability of bacterial relative to cyanobacterial prey. However, ratios of bacteria to cyanobacteria are largely invariable across the transect, as are ratios of protists to cyanobacteria. Both of these values would reasonably be expected to vary if responsible for the reported differences in $d$.

### 5. Conclusion

Our results demonstrate the distribution of microorganisms in the WTSP to be qualitatively similar to those reported for other highly-stratified oligotrophic regions. The entire transect length was characterized by a two-tier vertical niche partition, with *Prochlorococcus* and *Synechococcus* achieving abundance maxima in the $Z_u$, and PPE achieving abundance maxima in the $Z_l$, at depths coincident with the DCM. The strong relationships between $N_2$ fixation and primary producers demonstrates the central role of $N_2$ fixation in regulating ecosystem processes in the WTSP, with the influence of biologically fixed nitrogen being ex-

erted across all depths and across all classes of organisms in the study region. At the mesotrophic stations and LDB, increases in $N_2$ fixation rates are accompanied by increased production near the surface, and by increased abundances of *Synechococcus* relative to *Prochlorococcus*. At oligotrophic stations, the marked decrease in $N_2$ fixation rates is accompanied by greatly reduced phytoplankton abundances, which may translate directly into reduced proportions of HNA bacteria. This shift results in an increase in the importance of bottom-up controls in regulating the abundance of organisms in higher trophic levels. In the lower

euphotic zone, these changes may also influence the amount and the quality of nutrients available to PPE communities, influencing both the diversity and vertical distributions of the organisms they comprise.

### Acknowledgments

This is a contribution of the OUTPACE project (https://outpace.mio.univ-amu.fr/) funded by the French research national agency

(ANR-14-CE01-0007-01), the LEFE-CyBER program (CNRS-INSU), the GOPS program (IRD) and the CNES. We thank T. Moutin and S. Bonnet, chief scientists of the OUTPACE cruise. We are indebted to O. Grosso and S. Helias-Nunige for nutrient measurements, to S. Bonnet for $N_2$ fixation measurements, to J. Ras for HPLC pigment measurements, and to G. Rougier and M. Picheral for their help in CTD rosette management and data processing. We are grateful to the crew of the R/V L'Atalante for outstanding shipboard operation. National Science Foundation (NSF) OCE-1434916 award to S.D. supported S.D. and M.D;

NSF OCE-1458070 award to S.D. supported N.B.

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



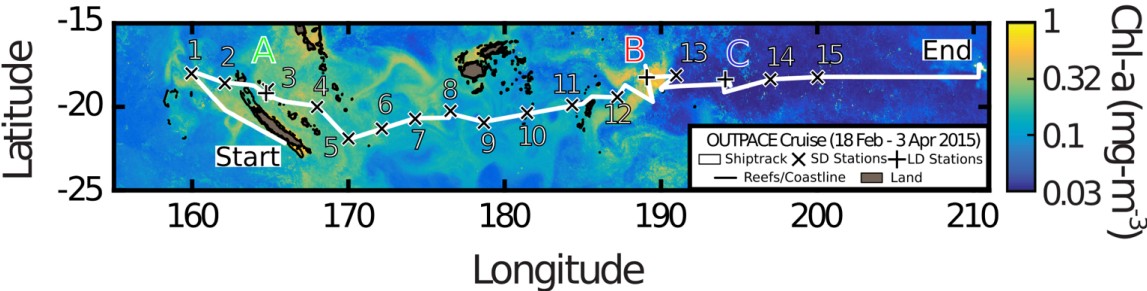

Figure 1: Quasi-Lagrangian surface Chlorophyll-a concentration (mg m$^{-3}$) in the sampling region. Data represent the
mean chlorophyll-a concentration March 2015. The white line identifies the track of the OUTPACE cruise, with the
sampled stations marked 1 to 15 (x) and the long duration stations marked A, B and C (+).





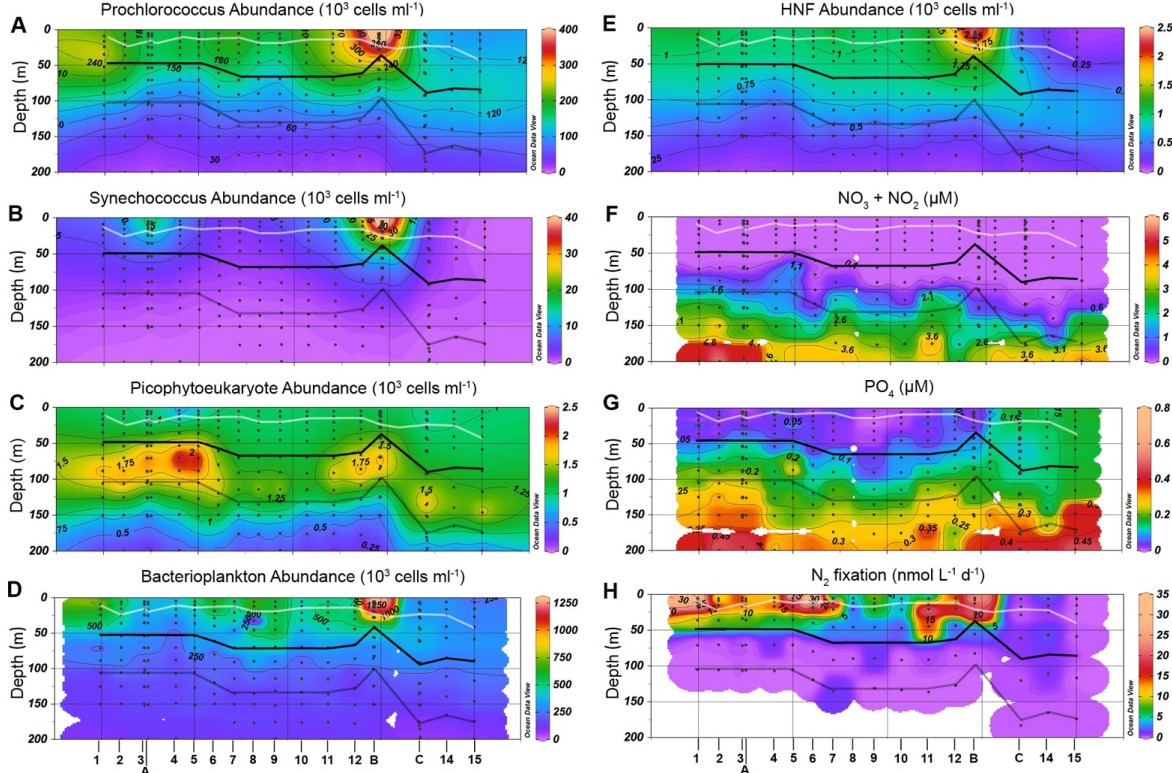

Figure 2: Contour plots depicting plankton abundances (A, B, C, D, E), nutrient concentrations (F, G) and N$_2$ fixation (H) distributions along the OUTPACE transect. White line represents the mixed layer depth. The black solid line represents the PAR$_{2.7}$

5 depth. Grey line represents PAR$_{0.1}$.



Figure 3: Abundance profiles for *Prochlorococcus* (A), *Synechococcus* (B), and PPE (C). Color-coded lines represent average
5   cell abundance, grey lines represent relative fluorescence for each group, based on cytometry data. Color-coded points represent
original observations for biogeochemical region (mesotrophic —Meso, LDB, and oligotrophic —Oligo), with shading represent-
ing standard error. Dotted and dashed horizontal lines represent average $PAR_{2.7}$ and $PAR_{0.1}$ depths, respectively.



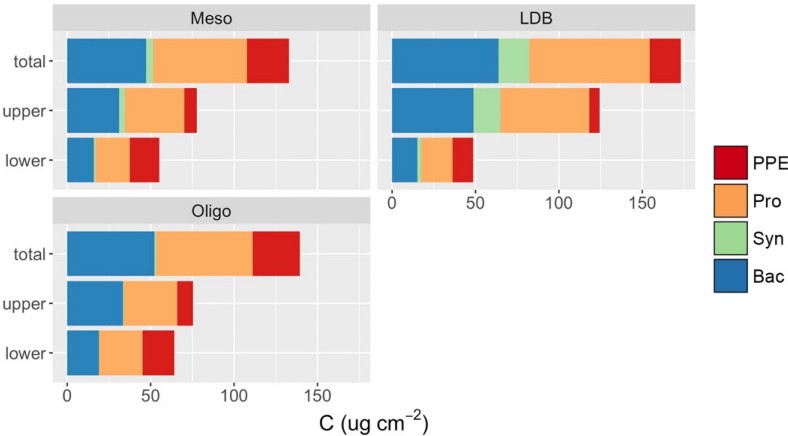

Figure 4: Contribution of different plankton groups (picophytoeukaryotes (PPE, red), *Prochlorococcus* (Pro, orange), *Synechococcus* (Syn, green), and heterotrophic bacteria (Bac, blue)) to depth-integrated biomass, averaged by biogeochemical region (mesotrophic —Meso, LDB, and oligotrophic —Oligo). Y axis corresponds to layer of euphotic zone, with "total" representing integrations from surface to $PAR_{0.1}$, "upper" representing integrations from surface to $PAR_{2.7}$, and "lower" referring to depths between $PAR_{2.7}$ and $PAR_{0.1}$.





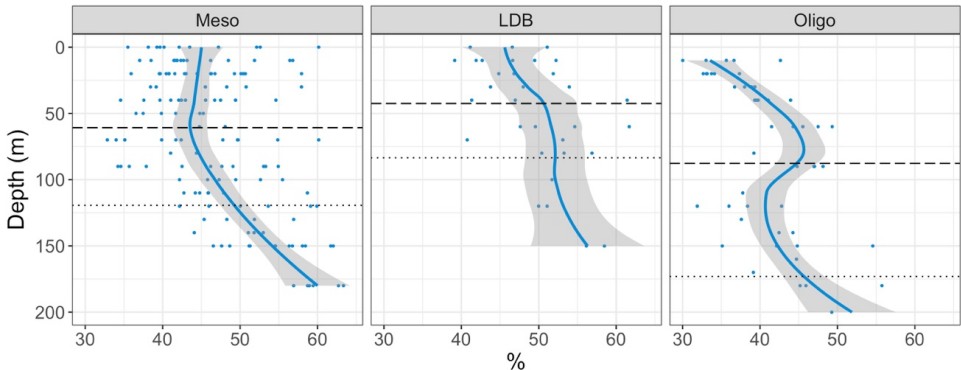

Figure 5: Percent of total bacteria accounted for by HNA, by biogeochemical region: mesotrophic —Meso, LDB, and oligo-
trophic —Oligo. Trendline calculated using LOESS regression. Shading represents standard error of samples at each depth.





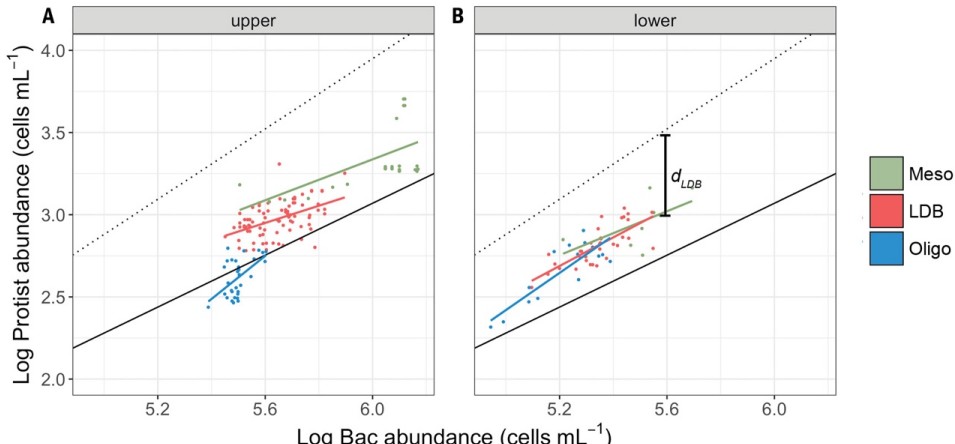

Figure 6: Log-Log plot of $Z_e$ bacteria and HNF abundances (A) and Log-Log plot of $Z_e$ bacteria and HNF abundances (B). Dots correspond to observed abundances, grouped by condition. Solid color-coded lines are regressions for abundance data corresponding to each biogeochemical condition. Solid black line corresponds to regression line for mean realized abundances in marine environments ($\log MRA = -1.67 + 0.97 * \log Bac$). The dotted line corresponds to theoretical maximum attainable abundance ($\log MAA = -2.47 + 1.07 * \log Bac$), as described in Gasol et al. (1994). $d_{LDB}$ included for clarification of $d$ parameter.




| cond. | zone | Pro. $10^{11}$ cells m$^{-2}$ | Syn. $10^{11}$ cells m$^{-2}$ | PPE $10^{11}$ cells m$^{-2}$ | Bacteria $10^{11}$ cells m$^{-2}$ | Protists $10^{11}$ cells m$^{-2}$ | NO$_x$* mmol m$^{-3}$ | PO$_4$ mmol m$^{-3}$ |
|---|---|---|---|---|---|---|---|---|
| *Meso* | $Z_u$ | 122 ± 31 | 3.42 ± 2.44 | 0.50 ± 0.10 | 271 ± 73 | 0.57 ± 0.10 | 0.02 ± 0.01 | 0.05 ± 0.04 |
| | $Z_l$ | 71 ± 24 | 0.79 ± 0.42 | 1.18 ± 0.33 | 139 ± 26 | 0.40 ± 0.09 | 0.43 ± 0.42 | 0.12 ± 0.05 |
| | $Z_e$ | 194 ± 52 | 4.21 ± 2.53 | 1.68 ± 0.36 | 410 ± 97 | 0.97 ± 0.18 | 0.22 ± 0.21 | 0.08 ± 0.03 |
| *LDB* | $Z_u$ | 183 ± 28 | 16.14 ± 8.64 | 0.42 ± 0.07 | 424 ± 108 | 0.91 ± 0.35 | 0.01 ± 0.01 | 0.03 ± 0.01 |
| | $Z_l$ | 65 ± 30 | 2.31 ± 1.13 | 0.81 ± 0.55 | 131 ± 40 | 0.36 ± 0.07 | 0.02 ± 0.01 | 0.07 ± 0.02 |
| | $Z_e$ | 248 ± 56 | 18.45 ± 7.68 | 1.24 ± 0.52 | 555 ± 141 | 1.27 ± 0.28 | 0.02 ± 0.02 | 0.05 ± 0.01 |
| *Oligo* | $Z_u$ | 110 ± 9 | 0.54 ± 0.20 | 0.63 ± 0.09 | 290 ± 32 | 0.41 ± 0.06 | 0.02 ± 0.01 | 0.13 ± 0.03 |
| | $Z_l$ | 89 ± 10 | 0.35 ± 0.06 | 1.26 ± 0.24 | 164 ± 17 | 0.43 ± 0.03 | 0.77 ± 0.42 | 0.20 ± 0.05 |
| | $Z_e$ | 199 ± 9 | 0.89 ± 0.20 | 1.89 ± 0.31 | 455 ± 30 | 0.85 ± 0.05 | 0.39 ± 0.20 | 0.17 ± 0.03 |

Table 1. Summary of depth-integrated abundances (*Prochlorococcus - pro.*, *Synechococcus - syn*, PPE, bacteria and protists) and of depth-averaged values of nutrient concentrations (NO$_X$ and PO$_4$), for different vertical zones ($Z_u$, $Z_l$ and Ze) and for individual biogeochemical conditions (cond.: mesotrophic –meso, LDB and oligotrophic –oligo). *NO$_2$ + NO$_3$



| cond. | zone | 19'BF mg m$^{-3}$ | 19'HF mg m$^{-3}$ | 19'BF/Chl a | 19'HF/Chl a | 19'HF/19'BF |
|-------|------|-------|-------|-------------|-------------|-------------|
| Meso | $Z_u$ | 0.005 ±0.007 | 0.021±0.014 | 0.037±0.037 | 0.156±0.060 | 5.925±2.634 |
|      | $Z_l$ | 0.040 ±0.021 | 0.057±0.031 | 0.171±0.058 | 0.240±0.041 | 1.616±0.704 |
| LDB  | $Z_u$ | 0.003 ±0.002 | 0.022±0.002 | 0.018±0.016 | 0.124±0.049 | 8.977±3.048 |
|      | $Z_l$ | 0.011 ±0.004 | 0.036±0.006 | 0.053±0.011 | 0.186±0.011 | 3.606±0.940 |
| Oligo | $Z_u$ | 0.003 ±0.003 | 0.012±0.007 | 0.064±0.018 | 0.254±0.026 | 4.246±1.220 |
|       | $Z_l$ | 0.024 ±0.010 | 0.037±0.010 | 0.170±0.051 | 0.271±0.051 | 1.712±0.502 |

Table 2. Average depth-integrated concentrations of 19'-hexanoyloxyfucoxanthin (19'HF), 19'-butanoyloxyfucoxanthin (19'BF).
Ratios of these pigments to Chl $a$ calculated using HPLC values for TChl $a$.





| cond. | area | chl a<br>mg m$^{-3}$ | pro<br>x 10$^{11}$ m$^{-3}$ | syn<br>x 10$^{11}$ m$^{-3}$ | PPE<br>x 10$^{11}$ m$^{-3}$ | %pro | Ref. |
|---|---|---|---|---|---|---|---|
| *Eu.* | Arabian Sea | 1.03 | 0.02 | 0.56 | 0.04 | 3 | Brown 1999 |
| *Eu.* | SE Pacific | 0.70 | 0.09 | 0.34 | 0.05 | 19 | Rii 2016b |
| *Eu.* | Arabian Sea | 0.61 | 0.54 | 0.43 | 0.12 | 30 | Brown 1999 |
| **LDB** | **SW Pacific** | **0.47** | **2.98** | **0.22** | **0.01** | **93** | **This study** |
| *Eu.* | NE Atlantic | 0.46 | 0.60 | 0.66 | 0.05 | 45 | Partensky 1996 |
| | | | | | | | |
| *Meso.* | NE Atlantic | 0.36 | 1.09 | 0.46 | 0.04 | 59 | Partensky 1996 |
| *Meso.* | SE Pacific | 0.27 | 1.21 | 0.21 | 0.07 | 81 | Rii 2016b |
| *Meso.* | Arabian Sea | 0.23 | 2.65 | 0.52 | 0.03 | 83 | Brown 1999 |
| ***Meso.*** | **SW Pacific** | **0.23** | **1.63** | **0.04** | **0.01** | **97** | **This study** |
| | | | | | | | |
| *Oligo.* | NE Atlantic | 0.17 | 1.06 | 0.01 | 0.01 | 97 | Partensky 1996 |
| *Oligo.* | N Pacific | 0.13 | 1.87 | 0.02 | 0.01 | 99 | Rii 2016a |
| *Oligo.* | SE Pacific | 0.13 | 0.74 | 0.01 | 0.02 | 96 | Rii 2016b |
| *Oligo.* | SE Pacific | 0.11 | 0.35 | 0.04 | 0.03 | 84 | Grob 2007 |
| ***Oligo.*** | **SW Pacific** | **0.10** | **1.16** | **0.01** | **0.01** | **99** | **This study** |

Table 3. Depth-averaged abundances for Chl *a*, *Prochlorococcus* (pro), *Synechococcus* (syn), and PPE across eutrophic (Eu),
mesotrophic (Meso), and oligotrophic (oligo) conditions. %pro calculated as *pro/(pro+syn+PPE)*. order to ensure that data were
comparable to our own, we only selected studies reporting both depth-integrated Chl *a* concentrations and the depth of integra-
tion used. In the case of Brown et al., integrations were performed from reported abundances. To standardize results, reported
abundances and chlorophyll concentrations were divided by the reported depths of integration (data not shown). OUTPACE val-
ues correspond to discrete fluorometric data collected at each station of the transect, with the exception of SD 1-3.

