# Peer review of "Microbial community structure in the Western Tropical South Pacific"

_Biogeosciences, 2017_

## Referee Comment (RC1) · Anonymous Referee #1 · 28 Mar 2018

General comments

This is an exhaustive and complete study in the West Tropical South Pacific Ocean, trying to elucidate which are the factors driving the microbial community structure in the photic layer (200 m). Authors did an amount of work measuring abundances of heterotrophic and phototrophic microorganisms, nutrients…etc, following a transect from mesotrophic to ultraoligotrophic area. Also, through established models they try to disentangle if microbes are top-down or bottom up controlled, and how is the role of limitation of nutrients and fixation of N2. I am conscious that to explain all this without the reader losing attention it is not easy. But sometimes, due to small inconsistencies that I detail below, makes that the flow of the story to be lost.

Specific comments

[Figure]

Abstract Page 1, line 23: What abundant are Prochlorococcus? Notice that Syne-chococcus are quantified.

Introduction and Material and Methods Page 2, line 22 the study was carried out from mesotrophy to ultraoligotrophy kinds of waters. In addition, in the introduction you are considering ultraoligotrophic areas and in the whole manuscript, you only refer to olig-otrophic. Page 3, line 34, I would move the definition of stations to field sampling section. Then all stations after LDB are oligotrophic including LDA and thereafter ul-traoligotrophic? Page 3, line 15, Vázquez-Dominguez is missing in the reference list Page 4, line 11, arrange sub-index for phosphate, nitrate, and nitrite

Results Page 4, line 37, Fig. 2 do not show temperature

Page 6, line 31, Why do you not show HNF and ciliate profiles?

Page 6, line 39, Gasol (1994) is missing in the reference list.

Page 6, Fig. 6. Very interesting the obtained results related with bottom-up and top-down issue. Although as the authors say there were no measures of grazing rates. Perhaps you might have a look on the paper of Lara et al. (2017) that for near stations to this study, during the MALASPINA cruise, they measured grazing and viral mortality rates (Lara, E., D. Vaqué, E.L. Sà, J.A. Boras, A. Gomes, ..., R. Massana, T.S. Catalá, G.M. Luna, S. Agustí, M. Estrada, J.M. Gasol & C.M. Duarte (2017) Unveiling the role and life strategies of viruses from the surface to the dark ocean Sci. Adv. 3: e1602565, doi: 10.1126/sciadv.1602565 3: 1-1).

Another paper from the same cruise that deals with this top-down, bottom-up issue from the point of view of bacteria could help you as is: Morán, X.A.G, J. M. Gasol, M. Pernice, J.-F. Mangot, R. Massana, E. Lara, D. Vaqué & C.M. Duarte (2017) Tempera-ture regulation on marine heterotrophic prokaryotes increases latitudinally as a breach between bottom-up and top-down controls Global Change Biol. 23:3956–3964. doi: 10.1111/gcb.13730 23: 3956-3964.

[Figure]

Page 7, line 5, About the interpretation of the large distance of d when using data from the Zl in the oligotrophic stations, since Prochlorococcus are pretty abundant, as well as Synechococcus and both could be a prey for HNF, I am wondering, which would be the result if you sum up all bacteria (heterotrophic and phototrophic) and apply the model again?. At the time that Gasol 1994, did the model it was very difficult to detect Prochlorococcus in the epifluorescence microscope, so, perhaps some of them were counted as heterotrophic bacteria after DAPI staining. Another way to corroborate this top-down bottom-up issue from the point of view of bacteria could be the application of Ducklow 1992 equation, relating bacterial biomass and bacterial production, but for that you will need bacterial production measurements. Did somebody measure it during this cruise? Ducklow H. 1992. Factors regulating Bottom-up control of bacterial biomass in open ocean plankton communities. Arch. Hydrobiol. Beih.Ergebn. Limnol. 37: 207-217

In summary, I consider this manuscript a relevant piece of work describing the microbial abundance and biomass through a gradient from meso to oligotrophic in a not explored tropical area of the Pacific Ocean, and I think that few changes will contribute to make it clearer.

---

## Referee Comment (RC2) · Anonymous Referee #2 · 30 Mar 2018

Journal: BG Title: Microbial community structure in the Western Tropical South Pacific Author(s): Nicholas Bock et al. MS No.: bg-2017-562 MS Type: Research article Special Issue: Interactions between planktonic organisms and biogeochemical cycles across trophic and N2 fixation gradients in the western tropical South Pacific Ocean: a multidisciplinary approach (OUTPACE experiment)

This MS describes picophytoplankton abundance in Western tropical south Pacific, where such information is strictly limited. I hope that my comments are helpful to improve the MS.

1) ANOVA results 1-a: Please show the summary table of Two-way ANOVA. Authors analyzed two-way ANOVA, but I am not sure two factors, and the aim of the analysis. If authors will show the summary table in the Result section and mention the strategy

of two-way ANOVA in Materials and Methods section, these are helpful for readers.

1-b: ANOVA assess the difference among the multiple assemblages (groups) and cannot determine which is higher than others. However, authors sometimes mentioned that one assemblage is significantly higher than others. For example, "values for %HNA were significantly greater at LDB relative to the mesotrophic and oligotrophic stations (ANOVA, p < 0.01)" in lines 6-7 of page 6. In the Materials and Method section, please describe the multiple comparison after ANOVA.

2) Bottom-up and top-down control of microbial communities Authors analyzed top-down or bottom-up control of microbial communities using counting data of various planktonic groups. In the analysis, author used heterotrophic bacterial counting data. However, I wonder that Prochlorococcus should also be included for this analysis, as the Prochlorococcus was numerically abundant, and the cell size is also overlapped with heterotrophic bacteria. Is there any data that heterotrophic flagellates grazed only on heterotrophic bacteria?

3) phagotrophy of PPE I think it is too much to say the possibility of phagotrophy of PPE under nitrate limited condition. Authors cannot make any concrete conclusion, I recommend that last two paragraphs (lines 16-32 of page 11) should be shorten.

4) NO2, NO3, PO4 Page 3, line 33, PO4, NO2, and NO3. Authors used PO4, NO2, NO3 in other sentences. Also, nitrate and nitrite are used instead of NO3 and NO2. Please correct them appropriately.

5) HNF abundance Authors mentioned that depth integrated abundances of HNF were greater in the Zu than the Zl at mesotrophic and oligotrophic stations (lines 15-16 of page 6). But, as far as I understood, HNF abundance in oligotrophic site is low in Zu (0.41) than that in Zl (0.43, Table 1).

6) Reference list Below papers are not included in the reference list;

Introduction Brum et al., 2015 Huang et al., 2015 Campbell and Vaulot, 1993 Karl,

1999 Denis et al., 2010 Blanchot and Rodier, 1996

Materials and methods Duhamel et al., 2014 Vazquez-Dominguez et al., 1999 Motoya et al., 1996 R core team, 2013

Results Gasol 1994

Discussion Herbland and Voituriez, 1979 Dore et al., 2008 Painter et al., 2014 Pérez et al., 2006 Van Wambeke et al., this issue Ducklow, 2002 Gasol, 1994

Below two papers, which are in the reference list, are not cited in the MS. Moutin et al., 2017 Tillson et al., 2004

7) Figures Fig. 3 are hard to see. It is preferred that axis color is changed to black, and size of labels is larger than the present. In my printing environment, axes of Fig. 4 are almost invisible.

---

## Author Response (AR1)

General comments

This is an exhaustive and complete study in the West Tropical South Pacific Ocean, trying to elucidate which are the factors driving the microbial community structure in the photic layer (200 m). Authors did an amount of work measuring abundances of heterotrophic and phototrophic microorganisms, nutrients. . .etc, following a transect from mesotrophic to ultraoligotrophic area. Also, through established models they try to disentangle if microbes are top-down or bottom up controlled, and how is the role of limitation of nutrients and fixation of N2. I am conscious that to explain all this without the reader losing attention it is not easy. But sometimes, due to small inconsistencies that I detail below, makes that the flow of the story to be lost.

**We would like to thank the reviewer for their constructive comments and suggestions, which have contributed to improve the manuscript. Please find below the reviewer's comments copied in italics and our response in regular font, with reference to changes in the manuscript in bold.**

Specific comments

*1) Abstract Page 1, line 23: What abundant are Prochlorococcus? Notice that Synechococcus are quantified.*

> Admittedly neglected to explicitly describe *Prochlorococcus* abundances. To incorporate this information, and in an effort to improve readability, we separated the description of physical characteristics describing "typical tropical structure" from the description of organism abundances and biomass. Lines 21 to 25 now read as follows:

> **At the most general level, we found a "typical tropical structure," characterized by a shallow mixed layer, a clear deep chlorophyll maximum at all sampling sites, and a deep nitracline. *Prochlorococcus* was especially abundant along the transect, accounting for 68 ± 10.6 % of depth-integrated phytoplankton biomass. Despite their relatively low abundances, picophytoeukaryotes (PPE) accounted for up 26 ± 11.6 % of depth-integrated phytoplankton biomass, while *Synechococcus* accounted for only 6 ± 6.9 %.**

*Introduction and Material and Methods*

*2) Page 2, line 22 the study was carried out from mesotrophy to ultraoligotrophy kinds of waters. In addition, in the introduction you are considering ultraoligotrophic areas and in the whole manuscript, you only refer to oligotrophic.*

> Criteria for mesotrophic/oligotrophic designation were based on those used in Grob 2011. This designation does indeed conflict with terminology used in the abstract, and in other manuscripts in

the special issue that are based on Moutin et al. (2017). To resolve this, we now designate mesotrophic stations as **Melanesian Archipelago (MA)**, and oligotrophic stations as **Gyre (GY)**, based on Moutin et al. (2017). The definition of stations now reads as follows:

**Following the designation used by Moutin et al. (2017), stations sampling the mesotrophic to oligotrophic waters of the Melanesian Archipelago (SD1 to SD12 and LDA) were classified as MA, while stations sampling the oligotrophic to ultra-oligotrophic waters of the western South Pacific Gyre (SD13 to SD15 and LDC) were classified as GY. Long duration station B was in the late stages of a phytoplankton bloom at the time of sampling (de Verneil et al., 2017). As such, it was analyzed independently of other stations on the transect and is simply referred to as LDB.**

*3) Page 3, line 34, I would move the definition of stations to field sampling section. Then all stations after LDB are oligotrophic including LDA and thereafter ultraoligotrophic?*

Moved definitions to field sampling sections, as suggested. The text from the above point now follows page 2, line 20.

*4) Page 3, line 15, Vázquez-Dominguez is missing in the reference list*

Thanks for highlighting. Added to list

*5) Page 4, line 11, arrange sub-index for phosphate, nitrate, and nitrite*

Thanks for highlighting. Corrected formatting to $NO_2$, $NO_3$, $PO_4$.

*6) Results Page 4, line 37, Fig. 2 do not show temperature*

The reference to figure 2 was intended to only apply to the mixed layer depth. We agree, though, that the reference is somewhat unclear (and the thermocline description perhaps extraneous). Lines 14 to 18 on page 4 now read as follows:

**Stations along the transect were characterized by warm sea surface temperatures (mean 29.4 ± 0.4 °C). The water column was strongly stratified along the entire transect, with mixed layer depths ranging from 21 ± 5 m for MA to 25 ± 8 m for GY (Fig. 2).**

*7) Page 6, line 31, Why do you not show HNF and ciliate profiles?*

HNF profiles were included with other plankton groups as Fig. 2E, and citation of Fig 2 E is now included in this sentence. Ciliate profiles were not included mainly because these profiles were previously published in Dolan (2016) and we only use the data in the discussion (i.e. not as a main result of the present study).

*8) Page 6, line 39, Gasol (1994) is missing in the reference list.*

Thanks for highlighting. Added to list

*9) Another way to corroborate this top-down bottom-up issue from the point of view of bacteria could be the application of Ducklow 1992 equation, relating bacterial biomass and bacterial production, but for that you will need bacterial production measurements. Did somebody measure it during this cruise? Ducklow H. 1992. Factors regulating Bottom-up control of bacterial biomass in open ocean plankton communities. Arch. Hydrobiol. Beih.Ergebn. Limnol. 37: 207-217*

France Van Wambeke did indeed measure bacterial production for the transect, and we did attempt to apply Ducklow's model to this data. The slopes of the model I regression were calculated on the log (BB) = f (log (BP)) relationship, based on BB in µg C $l^{-1}$, estimated from abundances based on a constant conversion factor for BP as used in our MS (i.e. 11.5 fg C per cell) and BP expressed in µgC $l^{-1}$ $h^{-1}$ and based on the leucine technique, using 1.5 kg C $mole^{-1}$. The slopes (± se) were 0.41 ± 0.030 for MA, 0.61 ± 0.049 for LDB, and 0.49 ± 0.071 for GY based on the maximum layer depth sampled (0-200 m). Biogeochemical condition was determined to have a significant effect on regression slopes (ANCOVA, $p < 0.01$). However, in $Z_u$, slopes were insignificant for MA, site LDB, and GY. In $Z_l$, slopes were insignificant in MA, 0.324 ± 0.015 in LDB

and 0.381 ± 0.116 at GY. By the criteria described in Ducklow 1992, this would correspond to "strong" bottom-up control at LDB, and "moderate" bottom-up control at MA and GY if we consider the 0-200m layer, but no, or weak bottom-up controls when focusing on $Z_u$ or $Z_l$ alone.

We added a sentence to address these results following line 15 on page 11. The new text reads as follows:

**Plotting bacterial abundances against the bacterial production data reported by Van Wambeke *et al.* (this issue, in press), and interpreting regression slopes using the criteria described by Ducklow *et al.* (1992) in the $Z_u$ layer, we found no evidence for bottom-up control of bacteria populations at MA, LDB, or GY.**

10) *Page 6, Fig. 6. Very interesting the obtained results related with bottom-up and top-down issue. Although as the authors say there were no measures of grazing rates. Perhaps you might have a look on the paper of Lara et al. (2017) that for near stations to this study, during the MALASPINA cruise, they measured grazing and viral mortality rates (Lara, E., D. Vaqué, E.L. Sà, J.A. Boras, A. Gomes, ..., R. Massana, T.S. Catalá, G.M. Luna, S. Agustí, M. Estrada, J.M. Gasol & C.M. Duarte (2017) Unveiling the role and life strategies of viruses from the surface to the dark ocean Sci. Adv. 3: e1602565, doi: 10.1126/sciadv.1602565 3: 1-1).*

11) *Another paper from the same cruise that deals with this top-down, bottom-up issue from the point of view of bacteria could help you as is: Morán, X.A.G, J. M. Gasol, M. Pernice, J.-F. Mangot, R. Massana, E. Lara, D. Vaqué & C.M. Duarte (2017) Temperature regulation on marine heterotrophic prokaryotes increases latitudinally as a breach between bottom-up and top-down controls Global Change Biol. 23:3956–3964. doi: 10.1111/gcb.13730 23: 3956-3964.*

Thanks for the very useful references. We added a sentence at line 33 on page 11 to acknowledge the importance of viruses in regulating bacterial abundance, citing the Lara *et al.* paper. The new text reads as follows:

**Finally, viruses undoubtedly contribute to the observed variation in bacterial abundances, with previous studies reporting viral lysis to be an equally, if not more important factor in controlling prokaryote mortality than grazing alone in the surface waters of the open ocean, with protistan grazing only becoming dominant in the DCM layer (Lara et al., 2017). Indeed, our *d* values were significantly reduced in $Z_l$ (including the DCM) compared to $Z_u$ in all the three regions investigated in the WTSP. However, there is a large degree of scatter along the 1:1 line in the relationship presented by Lara *et al.* (2017) between protist-mediated mortality and virus-mediated mortality, making it difficult to infer how viral lysis might have contributed to the reported differences in *d*.**

Reviewing at the Morán et al. paper, we also added the following after line 15, page 11 to account for the very low slopes that they reported in the South Pacific when using the Ducklow model. The new text reads, which immediately follows the text added in response to comment 9, reads as follows:

**This is similar to the results obtained by Morán *et al.* (2017), who applied the Ducklow model to data collected in the South Pacific during austral summer and reported very weak bottom-up control at all sampling sites, calculating regression slopes around 0.2 for samples between the surface and 4000 m. The authors likely would have obtained still lower slopes had their analysis been restricted to surface data alone, as we found for $Z_u$.**

12) *Page 7, line 5, About the interpretation of the large distance of d when using data from the Zl in the oligotrophic stations, since Prochlorococcus are pretty abundant, as well as Synechococcus and both could be a prey for HNF, I am wondering, which would be the result if you sum up all bacteria (heterotrophic and phototrophic) and apply the model again? At the time that Gasol 1994, did the model it was very difficult to detect Prochlorococcus in the epifluorescence microscope, so, perhaps some of them were counted as heterotrophic bacteria after DAPI staining.*

Redefining Bac as Bac+Pro+Syn did not seem to have any real impact on the model outputs. While both mean log(abundance) and $d$ values increased across all biogeochemical condition, upper euphotic zone ($Z_u$, where the bias to dim fluorescent *Prochlorococcus* cells is the greatest) values for $d$ remained significantly greater at GY compared to MA or LDB. Differences between MA and LDB remained insignificant. To address the concern, we added the following sentence at the end of the paragraph starting on line 25, page 11:

**To account for the possibility that *Prochlorococcus* cells were erroneously identified as heterotrophic bacteria at the time of the model's formulation, we repeated the analysis including abundances of *Prochlorococcus* on the x-axis. Although doing so increased $d$ values across all biogeochemical conditions, it did not qualitatively affect the relationship as described above.**

*In summary, I consider this manuscript a relevant piece of work describing the microbial abundance and biomass through a gradient from meso to oligotrophic in a not explored tropical area of the Pacific Ocean, and I think that few changes will contribute to make it clearer.*

[Figure]

Figure: Gasol model analysis repeated using abundances of heterotrophic bacteria, *prochlorococcus*, and *Synechococcus* rather than heterotrophic bacteria alone

*Biogeosciences Discuss.,*

https://doi.org/10.5194/bg-2017-562-RC2, 2018

[Figure]
Journal: BG Title: Microbial community structure in the Western Tropical South Pacific
Author(s): Nicholas Bock et al.
MS No.: bg-2017-562 MS Type: Research article
Special Issue: Interactions between planktonic organisms and biogeochemical cycles across trophic and N2 fixation gradients in the western tropical South Pacific Ocean: a multidisciplinary approach (OUTPACE experiment)

*This MS describes picophytoplankton abundance in Western tropical south Pacific, where such information is strictly limited. I hope that my comments are helpful to improve the MS.*

**We would like to thank the reviewer for their constructive comments and suggestions, which have contributed to improve the manuscript. Please find below the reviewer's comments copied in italics and our response in regular font, with reference to changes in the manuscript in bold.**

*1) ANOVA results 1-a: Please show the summary table of Two-way ANOVA. Authors analyzed two-way ANOVA, but I am not sure two factors, and the aim of the analysis. If authors will show the summary table in the Result section and mention the strategy of two-way ANOVA in Materials and Methods section, these are helpful for readers*

Summary table is included below and added to manuscript as Table 2. To clarify, the goal of the analysis was to quantify vertical and longitudinal differences in parameters measured along the transect. Categorical variables used in doing this were biogeochemical condition (mesotrophic, LDB, oligotrophic) and the region of euphotic zone (upper: $Z_u$ or lower, $Z_l$). Interpreting results using Tukey's HSD, two-way ANOVA was useful in separating between-condition differences in each section of the euphotic zone. Although the same could have been accomplished by subsetting the data to include only upper or lower euphotic zone samples and then performing one-way ANOVA on parameter values * condition, two-way ANOVA was a more straightforward approach.

*1-b: ANOVA assess the difference among the multiple assemblages (groups) and cannot determine which is higher than others. However, authors sometimes mentioned that one assemblage is significantly higher than others. For example, "values for %HNA were significantly greater at LDB relative to the mesotrophic and oligotrophic stations (ANOVA, p < 0.01)" in lines 6-7 of page 6. In the Materials and Method section, please describe the multiple comparison after ANOVA.*

Thanks for catching this! Comparisons were made using Tukey's Honest Significant Differences, by way of the R TukeyHSD package. To clarify, we added the following sentence to the methods:

**Tukey's Honest Significant Difference post-hoc test was used to compare group means when two-way ANOVA indicated significant between-group differences.**

*2) Bottom-up and top-down control of microbial communities Authors analyzed top-down or bottom-up control of microbial communities using counting data of various planktonic groups. In the analysis, author used heterotrophic bacterial counting data. However, I wonder that Prochlorococcus should also be included for this analysis, as the Prochlorococcus was numerically abundant, and the cell size is also overlapped with heterotrophic bacteria. Is there any data that heterotrophic flagellates grazed only on heterotrophic bacteria?*

One of the limitations of the Gasol model is that it assumes HNF to feed only on bacteria, which is very certainly not the case, as has been documented by a large number of studies. The sentence starting on line 29, page 11 is intended to address this, although perhaps does not go far enough. Regardless, adding *Prochlorococcus* to the analysis does indeed increase *d* values across all biogeochemical conditions, as shown in the figure below. However, values for *d* at the oligotrophic stations remain significantly greater than those at mesotrophic or bloom stations, and so the results are not affected qualitatively. Regardless, to clarify the likelihood of HNF feeding on cyanobacteria, we modified the sentence starting on line 29, page 11 to read as follows:

**Given that the HNF abundances predicted by the model are calculated on the assumption that HNF only graze on heterotrophic bacteria, the increase in *d* at GY may reflect increased grazing on cyanobacterial prey. Indeed, previous studies have reported HNF to graze on cyanobacteria, generally at rates similar to those reported for grazing on heterotrophic bacteria (Christaki, 2001; Cuevas and Morales, 2006; Ferrier-Pagès and Gattuso, 1998).**

Also added the following sentence following at the end of the paragraph starting on line 25, page 11:

**Moreover, including abundances of both heterotrophic and autotrophic bacteria when calculating values for *d* does not qualitatively affect the comparison between sites or layers as described above.**

*3) phagotrophy of PPE I think it is too much to say the possibility of phagotrophy of PPE under nitrate limited condition. Authors cannot make any concrete conclusion, I recommend that last two paragraphs (lines 16-32 of page 11) should be shorten.*

Second to last paragraph (lines 16-32 of page 11) was shortened as recommended. Paragraph (which was also combined with final paragraph) now reads as follows:

**By reducing bacterial abundances relative to those of HNF, the reduction of *d* reported in $Z_l$ may also result from phagotrophy by PPE. Feeding experiments in the North Atlantic have demonstrated small plastidic eukaryotes to account for up to 90 % of bacterivory in nutrient-limited waters (Zubkov and Tarran, 2008), while laboratory and field studies have demonstrated increased feeding rates specifically in response to P limitation (Christaki et al., 1999; McKie-Krisberg et al., 2015).**

*4) NO2, NO3, PO4 Page 3, line 33, PO4, NO2, and NO3. Authors used PO4, NO2, NO3 in other sentences. Also, nitrate and nitrite are used instead of NO3 and NO2. Please correct them appropriately.*

Thanks for highlighting. Fixed subscript on page 3, line 33 and replaced nitrate and nitrite as needed.

*5) HNF abundance Authors mentioned that depth integrated abundances of HNF were greater in the Zu than the Zl at mesotrophic and oligotrophic stations (lines 15-16 of page 6). But, as far as I understood, HNF abundance in oligotrophic site is low in Zu (0.41) than that in Zl (0.43, Table 1)*

This was admittedly an oversight. We rewrote the sentence to more accurately reflect values provided in Table 1, and repeated analysis using ANOVA for consistency with other results. Lines 15-17 of page 6 now read as follows:

**Depth integrated abundances were significantly greater in $Z_u$ than $Z_l$ at MA and GY (ANOVA, $p < 0.01$), while there was no significant change in HNF abundances with depth at GY.**

*6) Reference list Below papers are not included in the reference list;*

Thanks for highlighting. All indicated papers added to reference list

*7) Below two papers, which are in the reference list, are not cited in the MS. Moutin et al., 2017 Tillson et al., 2004*

Thanks for highlighting. Papers removed from reference list

*8) Figures Fig. 3 are hard to see. It is preferred that axis color is changed to black, and size of labels is larger than the present. In my printing environment, axes of Fig. 4 are almost invisible.*

Thanks for highlighting. We will thicken axes on figures 3 and 4 and will increase label size on figure 3 to match that of other figures in the manuscript.

[Figure]

Figure: Gasol model analysis repeated using abundances of both heterotrophic bacteria and *prochlorococcus* rather than heterotrophic bacteria alone

| | DF | Pro | | Syn | | PPE | | Bac | | %HNA | | HNF | | NOx | | PO4 | |
|---|---|---|---|---|---|---|---|---|---|---|---|---|---|---|---|---|---|
| | | F | p | F | p | F | p | F | p | F | p | F | p | F | p | F | p |
| Euphotic layer | 1 | 49.23 | < 0.01 | 23.35 | < 0.01 | 68.08 | < 0.01 | 88.29 | < 0.01 | 5.36 | 0.021 | 24.66 | < 0.01 | 42.38 | < 0.01 | 40.6 | < 0.01 |
| Area | 2 | 2.94 | 0.06 | 25.88 | < 0.01 | 3.15 | 0.054 | 4.57 | 0.017 | 16.90 | < 0.01 | 6.041 | 0.005 | 12.62 | < 0.01 | 113.2 | < 0.01 |
| Interaction | 2 | 7.03 | < 0.01 | 15.78 | < 0.01 | 0.85 | 0.436 | 5.97 | 0.005 | 0.69 | 0.501 | 11.31 | <0.01 | 11.88 | < 0.01 | 3.01 | 0.057 |

Table: Summary table of two-way ANOVA results for parameters analyzed in this study. Row 1 (euphotic layer) tests for significant differences between mean parameter values across different layers of the euphotic zone ($Z_u$ vs $Z_l$). Row 2 (condition) tests for significant differences between mean parameter values across different biogeochemical areas (MA vs LDB vs GY) on mean parameter values. Row 3 (interaction) tests for combined effect of euphotic layer and biogeochemical condition on mean parameter values. Relationships for Pro, Bac, HNF, NOx, and PO4 calculated from depth-integrated abundances; Relationships for %HNA calculated from raw values.

Changes to bg-2017-562 manuscript and abstract in response to associate editor report and reviewer comments

- o Changes to abstract
    - ▪ Lines 10 - 14
        - • In response to comment 1 by anonymous referee #1, text was revised to read as follows: **"At the most general level, we found a "typical tropical structure," characterized by a shallow mixed layer, a clear deep chlorophyll maximum at all sampling sites, and a deep nitracline.** *Prochlorococcus* **was especially abundant along the transect, accounting for 68 ± 10.6 % of depth-integrated phytoplankton biomass. Despite their relatively low abundances, picophytoeukaryotes (PPE) accounted for up 26 ± 11.6 % of depth-integrated phytoplankton biomass, while Synechococcus accounted for only 6 ± 6.9 %"**
- o Changes to manuscript text
    - ▪ Throughout manuscript
        - • Minor changes for formatting and grammar
        - • In response to comment 2 by anonymous referee #1, station groupings "oligotrophic" and "mesotrophic" replaced with "MA" and "GY" throughout document.
        - • Citations of other manuscripts appearing in special issue 894 updated as necessary to reflect changes in publication status.
        - • Changes made to in-text references to tables, reflecting change in table numbering
        - • "Protists" replaced with the more accurate term "heteronanoflagellate (HNF)" throughout text.
    - ▪ Page 1
        - • Lines 30 - 32.
            - o Corrected misleading and inappropriately cited statement that " Oligotrophic oceans, covering approximately 40 percent of the earth's surface, represent one of the earth's largest biomes. Despite relatively reduced productivity compared to coastal and upwelling regions, these nutrient-limited ecosystems account for an 15 estimated 90 percent of global marine primary production (Karl et al., 2012)." Text now reads: **"Subtropical oligotrophic oceans, covering approximately 40 percent of the earth's surface, represent one of the earth's largest biomes (Sarmiento et al., 2004). Despite relatively reduced productivity compared to coastal regions, these nutrient-limited ecosystems are estimated to account for one-quarter of annual net marine primary production (Field, 1998)."**
    - ▪ Page 2
        - • Lines 18-19
            - o Added reference to recent paper (Tenório et al., 2018) reporting on community structure in Southwest Pacific. Changed "one" to "two" in line 34 to reflect this addition.
    - ▪ Page 3
        - • Lines 3-6
            - o In response to comment 2 by anonymous referee #1, based grouping of sampling sites on designation used by Moutin et al. (2017). Text on page 3, lines 15-19 of the previously submitted version were changed to the following: **"In keeping with Moutin et al. (2017), stations sampling the waters of the Melanesian Archipelago (SD1 to SD12 and LDA) were classified as MA, while stations sampling the western South Pacific Gyre (SD13 to SD15 and LDC) were classified as GY. Long duration station B was in the late stages of a phytoplankton bloom at the time of sampling (de Verneil et al.,**

**2018). As such, the station was analyzed independently of other stations on the transect, and is simply referred to as LDB."**

- Also in response to comment 3 by anonymous referee #1, above text was moved to page 3, lines 3-6 of submitted version.

- Page 4
  - Line 15
    - In response to comment 5 by anonymous referee #1 and comment 4 by anonymous referee #2, "PO4, NO2, and NO3" were reformatted to "$PO_4$, $NO_2$, and $NO_3$."
  - Lines 26-27
    - To clarify that use of multiple samples from the same station did not violate assumptions of independence when conducting statistical analyses, added the following sentence: **" Because each cast was made on a different day of occupation, doing so did not violate assumptions of independence during subsequent statistical analyses."**
  - Lines 29-30
    - In response to comment 1b by anonymous referee #2, use of post-hoc statistical tests was elaborated upon with the following text: **"Tukey's Honest Significant Difference post-hoc test was used to compare group means when two-way ANOVA indicated significant between-group differences."**

- Page 5
  - Lines 1-2
    - In response to comment 6 by anonymous referee #1, page 4, lines 14-18 of the previously submitted manuscript were revised for clarity. Revised text reads as follows: **"Stations along the transect were characterized by warm sea surface temperatures (mean 29.4 ± 0.4 °C). The water column was strongly stratified along the entire transect, with mixed layer depths ranging from 21 ± 5 m for MA to 25 ± 8 m for GY (Fig. 2)."**
  - Lines 5-6
    - Added units ($m^{-1}$) for light attenuation coefficient
  - Line 17
    - In response to comment 4 by anonymous referee #2, "Nitrate plus nitrite ($NO_X$)" on page 4, line 32 of original manuscript was revised to **"$NO_X$"**

- Page 6
  - Line 5
    - Revised text describing Gasol model for clarity, revising **"Specifically, this approach compares observed ratios of bacteria to HNF with HNF abundance maxima estimated a Lotka-Volterra model"** to read **"Specifically, this approach compares observed ratios of bacteria to HNF with HNF abundance maxima estimated from empirical data and theoretical interactions between bacteria and HNF."**
  - Lines 35-36
    - In response to comment 5 by anonymous referee #2, lines 15-18 were revised for clarity and for consistency with rest of manuscript. Revised text reads as follows: **"Depth integrated abundances of HNF were significantly greater in $Z_u$ than $Z_l$ at MA and GY (ANOVA, p < 0.01), while there was no significant change in HNF abundances with depth at LDB."**

- Page 8
  - Line 20

- o Removed first two sentences of paragraph to avoid redundant definitions of prevailing biogeochemical conditions.
  - Lines 35 - 37
    - o In order to align discussion with additional data included in Table 4, lines 33 - 37 were revised to read as follows: **" To compare our results to those from other ocean basins, we conducted a meta-analysis of datasets reporting *Prochlorococcus*, *Synechococcus*, and PPE abundances alongside Chl *a* concentrations. Mean depth-integrated abundances of *Synechococcus* and PPE measured along the OUTPACE transect were similar to those reported elsewhere, as were those of *Prochlorococcus* at MA and GY (Table 4). However, mean depth-integrated abundances of *Prochlorococcus* at LDB were considerably higher than mean values for other regions."**
- Page 9
  - Lines 16-19
    - o Text from lines 22 to 28, page 8 of the previous draft moved to page 9, lines 16 - 19 of the submitted draft, and revised to read as follows: **"The high relative abundance of *Prochlorococcus* at LDB compared to sites with similar Chl *a* concentrations may be due to having captured the bloom in decay (de Verneil., 2017), with nutrients having been largely depleted at the time of measurement and relative *Prochlorococcus* abundances returning to levels more representative of the WTSP. Alternatively, the bloom conditions may be incomparable to regions with more persistent inputs of nutrients."**
  - Lines 35 - 36
    - o Following text added at suggestion of co-author: **" while Caffin et al. have demonstrated the efficient transfer of N fixed by UCYN-B bacteria to the planktonic food web along the OUTPACE transect (2018)."**
- Page 11
  - Line 19
    - o Text added to clarify structure of argument. First sentence of page 10, line 32 now reads **" There are several possible explanations for this result."**
  - Lines 5-9
    - o To address comment 11 of anonymous referee #1, following text was added: **"This is similar to the results obtained by Morán *et al*. (2017), who applied the Ducklow model to data collected in the South Pacific during austral summer and reported very weak bottom-up control at all sampling sites, calculating regression slopes around 0.2 for samples between the surface and 4000 m. The authors likely would have obtained still lower slopes had their analysis been restricted to surface data alone, as we found for $Z_u$."**
  - Lines 36-39; Page 12, Lines 1-2
    - o In response to comment 2 by anonymous referee #2, limitations of Gasol model addressed more explicitly, with additional references cited as appropriate. Text from page 11 line 29-32 revised to read **" The increase in *d* at GY could alternatively result from increased grazing on cyanobacterial prey, given that the HNF abundances predicted by the Gasol model are calculated on the assumption that HNF only graze on heterotrophic bacteria. Previous studies have reported HNF to graze on cyanobacteria, generally at rates similar to those reported for grazing on heterotrophic bacteria (Christaki,**

**2001; Cuevas and Morales, 2006; Ferrier-Pagès and Gattuso, 1998). However, ratios of bacteria to cyanobacteria are largely invariable across the transect, as are ratios of HNF to cyanobacteria. Both of these values would reasonably be expected to vary if responsible for the reported differences in *d*.**

- o Above text was moved to page 11, lines 34-39 of submitted version of manuscript.
- Page 12
  - o Lines 2-5
    - ▪ In response to comment 12 by anonymous referee #1 and comment 2 by anonymous referee #2, following text was added: **"To account for the possibility that Prochlorococcus cells were erroneously identified as heterotrophic bacteria at the time of the model's formulation, we repeated the analysis including abundances of Prochlorococcus on the x-axis. Although doing so increased d values across all biogeochemical conditions, it did not qualitatively affect the relationship as described above (data not shown)."**
  - o Lines 6-9
    - ▪ In response to comment 3 by anonymous referee #2, lines 16-32 on page 11 of the previously submitted draft were revised to read as follows: **"The reduction of *d* reported in $Z_l$ may result from phagotrophy by PPE, by reducing bacterial abundances relative to those of HNF. Feeding experiments in the North Atlantic have demonstrated small plastidic eukaryotes to account for up to 90 % of bacterivory in nutrient-limited waters (Zubkov and Tarran, 2008), while laboratory and field studies have demonstrated increased feeding rates specifically in response to P limitation (Christaki et al., 1999; McKie-Krisberg et al., 2015)."**
  - o Lines 14-19
    - ▪ To address comment 11 by anonymous referee #1, following text was added: **"Finally, viruses undoubtedly contribute to the observed variation in bacterial abundances, with previous studies reporting viral lysis to be an equally, if not more important factor in controlling prokaryote mortality than grazing alone in the surface waters in open ocean, with protistan grazing only becoming dominant in the DCM layer (Lara et al., 2017). Indeed, *d* values were smaller in $Z_l$ (including the DCM) than in $Z_u$ in all three regions investigated in the WTSP. However, the relationship presented by Lara et al (2017) between protist-mediated mortality and virus-mediated mortality is very large along the 1:1 line, making it difficult to infer how viral lysis might have contributed to the reported differences in *d*."**
  - o Line 30
    - ▪ Text revised for clarity/accuracy. Revised text reads **"This shift is coincident with"** in the place of **"This shift results in."**
  - o Lines 30-31
    - ▪ Text revised for clarity/accuracy. Revised text reads **"reduction in the importance of top down controls in regulating bacteria abundance under nutrient limited conditions"** in the place of **"increase in the importance of**

**bottom up controls in regulating the abundance of organisms in higher trophic levels."**

- Page 13
  - Lines 5-6
    - Abbreviated first names in acknowledgments were changed to full spelling
  - Lines 9-11
  - To acknowledge the use of data in conducting meta-analysis of FCM data, the following text was added: **"This study uses data from the Atlantic Meridional Transect Consortium (NER/0/5/2001/00680), provided by the British Oceanographic Data Centre and supported by the Natural Environment Research Council."**
- Pages 14-19
  - To address comments 6 and 7 by anonymous referee #2, indicated references were either added or removed.
  - References to other manuscripts appearing in special issue 894 were modified to reflect updates in their publication status.
- Page 22
  - Colors used in figure 3 modified for consistency with other figures
  - Labels scaled for consistency with other figures
  - "Meso" and "Oligo" labels in legend replaced with "MA" and "GY" for consistency with other figures and with manuscript text.
  - Top x-axis labels modified to reduce cluttering and to ensure legibility
  - In response to comment 8 by anonymous referee #2, plot boundaries thickened to ensure visibility
- Page 23
  - In response to comment 8 by anonymous referee #2, plot boundaries thickened to ensure visibility
  - "u" replaced with Greek mu in x axis label
  - Legend beneath LDB plot to reduce image dimensions
  - "Meso" and "Oligo" labels in legend replaced with "MA" and "GY" for consistency with other figures and with manuscript text.
  - Labels scaled for consistency with other figures
  - Ordering of labels in figure 4 caption modified to better reflect order in figure. Revised text reads as follows: "**(heterotrophic bacteria (Bac, blue),** *Synechococcus* **(Syn, green),** *Prochlorococcus* **(Pro, organge), and picophytoeukaryotes (PPE, red))"**
- Page 24
  - "Meso" and "Oligo" labels in legend replaced with "MA" and "GY" for consistency with other figures and with manuscript text.
  - Dotted lines added between rows to aid in legibility
  - Dotted lines added between rows to aid in legibility
  - Labels scaled for consistency with other figures
- Page 25
  - "Meso" and "Oligo" labels in legend replaced with "MA" and "GY" for consistency with other figures and with manuscript text.
  - Dotted lines added between rows to aid in legibility
  - Labels scaled for consistency with other figures
- Page 26
  - Table caption revised for clarity. Revised text reads as follows: **"Summary of depth-integrated abundances for** *Prochlorococcus* **(Pro),** *Synechococcus* **(Syn), picophytoeukaryotes (PPE) and for depth-averaged values of nutrient concentrations ($NO_X$ and $PO_4$), for different vertical zones ($Z_u$, $Z_l$ and Ze) and for individual**

**biogeochemical conditions (cond.: MA, LDB, and GY). $^{*}NO_2 + NO_3$"**
- o Dotted lines added between rows to aid in legibility
- Page 27
  - o In response to comment 1 by anonymous referee 2, summary table of two-way ANOVA was added to manuscript as "Table 2."
  - o Dotted lines added between rows to aid in legibility
- Page 28
  - o Table 2 renumbered to Table 3
  - o Reduced decimal places in column 7
- Page 29
  - o Table 3 renumbered to Table 4
  - o Additional data incorporated into table.
  - o References to other publications replaced with references to respective data sources
  - o "n" column added to account for differences in number of sampling sites used in summary statistics.
  - o Dotted lines added between rows to aid in legibility
- o Changes to manuscript formatting
  - ▪ All major text components (title, authors, headings, subheadings, etc.) were adjusted to match the Copernicus template
  - ▪ References formatting adjusted to uniform line spacing of 1.2, with 7 points following each entry
  - ▪ All figures justified left
  - ▪ Figure 2
    - Labels scaled for consistency with other figures
    - Prochlorococcus and Synechococcus italicized in Fig 2a and Fig 2b
  - ▪ Figure 3
    - Labels scaled for consistency with other figures
    - Breaks on top x axes reformatted to reduce overlap and to ensure consistency between Fig3 a-c
    - In response to comment 8 by anonymous referee #2, plot borders thickened to ensure visibility
  - ▪ Figure 4
    - Ordering of labels
  - ▪ Figure 5
    - Labels scaled for consistency with other figures
  - ▪ Figure 6
    - Labels scaled for consistency with other figures
    - Equations scaled

[revised manuscript text omitted]

**Table 4: Summary table of two-way ANOVA results for parameters analyzed in this study. Row 1 (euphotic layer) tests for significant differences between mean parameter values across different layers of the euphotic layer ($Z_u$ or $Z_l$) on mean parameter values. Row 2 (condition) tests for significant differences between mean parameter values across different biogeochemical area (MA, LDB, and GY). Row 3 (interaction) tests for differences between mean parameter values across both euphotic layer and biogeochemical condition. Relationships for Pro, Bac, HNF, $NO_x$, and $PO_4$ calculated from depth-integrated abundances; Relationships for %HNA calculated from raw values.**